# Structure-guided design of endosomolytic chloroquine-like lipid nanoparticles for mRNA delivery and genome editing

Zhen Liu[1,4], Jiacai Wu[2,4], Ning Wang[2], Yongqi Lin[2], Ruiteng Song[1], Min Zhang [3] & Bin Li [1,2] ✉

Despite remarkable progress in designing RNA delivery systems, endosomal escape remains a recognized challenge for efficient RNA delivery. In this study, we develop a robust mRNA delivery platform termed endosomolytic chloroquine-like optimized lipid nanoparticles (ecoLNPs) for versatile mRNA delivery in vitro and in vivo via integrating the signature scaffold extracted from endosomolytic chloroquine into ionizable lipids. RNase-resistant ecoLNPs are capable of delivering a broad variety of mRNA payloads to diverse cell types, even hard-to-transfect 3D cells, with an efficiency of up to 18.9-fold higher than that of commercial transfection reagents. The pH-responsive endosomolytic activity of ecoLNPs can be largely attributed to the proton sponge effect and saposin B-promoted membrane disruption. In vivo, ecoLNPs enable potent local and systemic mRNA delivery and exhibit comparable potency to the clinically approved mRNA vaccine carrier, but strong tropism for lymph nodes following intramuscular injection. Furthermore, ecoLNPs are able to retain in vivo delivery potency for at least one week under non-frozen conditions and induce efficient genome editing in transgenic mice. Overall, the structure-guided integration strategy provides a pathway for de novo design of endosomolytic mRNA delivery systems.

The past few years have witnessed a great breakthrough in mRNA nanomedicine for the prevention of coronavirus disease 2019 (COVID-19)[1,2]. As a new therapeutic modality, mRNA-based therapeutics hold tremendous potential in gene and protein replacement therapy, cancer immunotherapy, and gene editing beyond mRNA vaccines[3–9]. The high degree of designability and rapid manufacturing process allow for flexible tailoring and production of mRNA molecules at an unparalleled speed[10]. The therapeutic effect of in vitro-transcribed mRNA, however, was hindered by its intrinsic instability and immunogenicity[10]. To achieve efficient cytoplasmic delivery, continuous efforts have been made in recent decades in mRNA engineering and mRNA delivery systems[6,7]. The former such as mRNA chemical modification and codon optimization contributes to

improved stability and reduced immunogenicity[6,7,11]. mRNA delivery systems such as lipid nanoparticles (LNPs), polymeric nanoparticles, and inorganic nanoparticles are capable of protecting mRNA payloads from degradation and facilitating their cellular uptake and endosomal escape, thereby resulting in significant improvements in both stability and translation efficiency[3–9].

LNPs initially used for siRNA delivery represent the most clinically advanced nucleic acid delivery system with high encapsulation efficiency, reproducibility and scalability, making them attractive non-viral vehicles for mRNA delivery[9,12–14]. Typical LNPs consist of four components: ionizable lipids, helper lipids, cholesterol, and PEGylated lipids, among which electrostatic interactions between ionizable lipids and negatively charged mRNA molecules are the main forces driving

[1]Department of Infectious Disease, Shenzhen People's Hospital, The Second Clinical Medical College, Jinan University, Shenzhen, China. [2]School of Medicine, Southern University of Science and Technology, Shenzhen, China. [3]Department of Ophthalmology, Shenzhen People's Hospital, The Second Clinical Medical College, Jinan University, Shenzhen, China. [4]These authors contributed equally: Zhen Liu, Jiacai Wu. ✉e-mail: libin@mail.sustech.edu.cn

their self-assembly[5,8,12–14]. Upon internalization by target cells, endosomal escape becomes the rate-determining step and key determinant of LNP potency, posing an enormous delivery challenge for mRNA nanomedicine[15–17]. The endosomal escape efficiency of siRNA-LNP has been reported to be as low as 2% even for one of the most advanced LNPs[18]. In recent years, a number of strategies aimed at overcoming endosomal entrapment such as the development of acid-degradable LNPs, tuning LNP compositions and fusogenicity, as well as targeting recycling endosomes with small molecules have been developed to maximize protein output[16,19,20], which is especially important for therapeutic mRNA that require higher levels of protein expression than prophylactic mRNA products to achieve therapeutic purposes[6].

Chloroquine is a classical anti-malarial drug that has been used worldwide for several decades and remains on the latest World Health Organization Model List of Essential Medicines as a first-line treatment for chloroquine-sensitive *Plasmodium vivax* malaria in many countries[21,22]. In addition to malaria, chloroquine has been used widely in current clinical practice for the treatment of rheumatic diseases such as rheumatoid arthritis and systemic lupus erythematosus[23]. Other potential applications under investigation include antiviral (e.g. HIV and SARS-CoV-2) and anticancer treatment[24]. Structurally, chloroquine belongs to a tertiary amine weak base that becomes protonated and entrapped in membrane-enclosed low-pH organelles due to low retro-diffusion rate. It has been utilized as a nucleic acid delivery enhancer for several decades, but its function has only been observed in in vitro experiments in the presence of delivery carriers[25–27]. Thus far, several mechanisms of action have been proposed to explain its enhancing property[23,27]. One possible explanation is that chloroquine can destabilize or fuse with endo-lysosomal membranes, thus

promoting the release of co-delivered molecules. Other proposed mechanisms include pore formation and pH-buffering effects[23,27,28].

In this work, we develop a strategy combining signature structural features extracted from chloroquine and ionizable lipids to create an endosomolytic chloroquine-like lipid. We thus construct a library of chloroquine-like lipids (CIIs) built from combinations of three building blocks (Fig. 1). After screening and optimization, we evaluate the top-performing formulation, CF3-2N6-UC18 ecoLNPs, from multiple aspects to demonstrate their robust mRNA delivery efficiency in vitro and in vivo (Fig. 1). We also elucidate the endocytosis pathways required for the cellular uptake of ecoLNPs and the potential mechanisms of action for their endosomolytic activity by designing a control lipid bearing the identical linker and tails to the lead lipid but with a non-quinoline scaffold. After testing the routes of administration, range of dosage required for systemic mRNA delivery, and expression kinetics of ecoLNPs in wild-type mice, we ultimately assess in vivo genome editing mediated by ecoLNPs in transgenic mice by delivering both Cre mRNA and CRISPR-Cas9 mRNA, further validating the biomedical potential of ecoLNPs.

## Results
### Rational design and screen of CIIs
Inspired by the ability of chloroquine to enhance gene transfection in vitro, we hypothesized that chemical entities with both quinoline and ionizable hydrophobic aliphatic hydrocarbon moieties could be leveraged as endosomolytic vehicles for efficient mRNA delivery. To achieve modular design of CIIs, the chemical structure of parental chloroquine was initially divided into three segments, namely a 7-substituted quinoline ring as a scaffold, two alkyl groups attached to

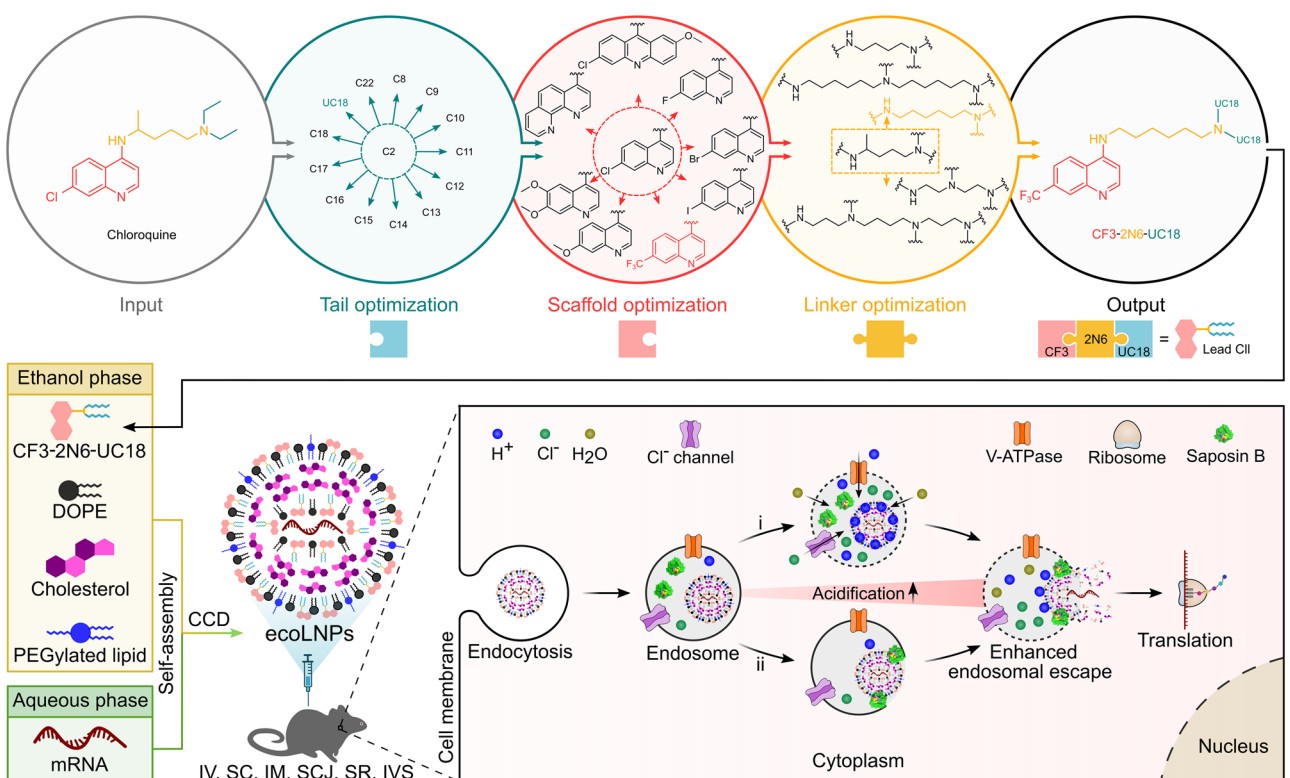

**Fig. 1 | Schematic illustration of structure-guided design of ecoLNPs for mRNA delivery and genome editing.** A library of CIIs was constructed by combining three distinctive modules (the scaffold, linker, and tail) extracted from chloroquine and ionizable lipids. The lead CII (CF3-2N6-UC18) identified from the library, together with auxiliary lipids (DOPE, cholesterol, and PEGylated lipids) spontaneously self-assembles into ecoLNPs in the presence of mRNA molecules after formulation optimization using CCD. ecoLNPs administered through multiple administration routes show potent mRNA delivery in vivo. The proposed mechanism of action of ecoLNPs as an endosomolytic mRNA delivery vehicle involves (i) the proton sponge effect and (ii) saposin B (sapB)-promoted membrane disruption. DOPE 1,2-dioleoyl-*sn*-glycero-3-phosphoethanolamine, CCD Central Composite Design, IV intravenous injection, SC subcutaneous injection, IM intramuscular injection, SCJ subconjunctival injection, SR subretinal injection, IVS intravesical instillation.

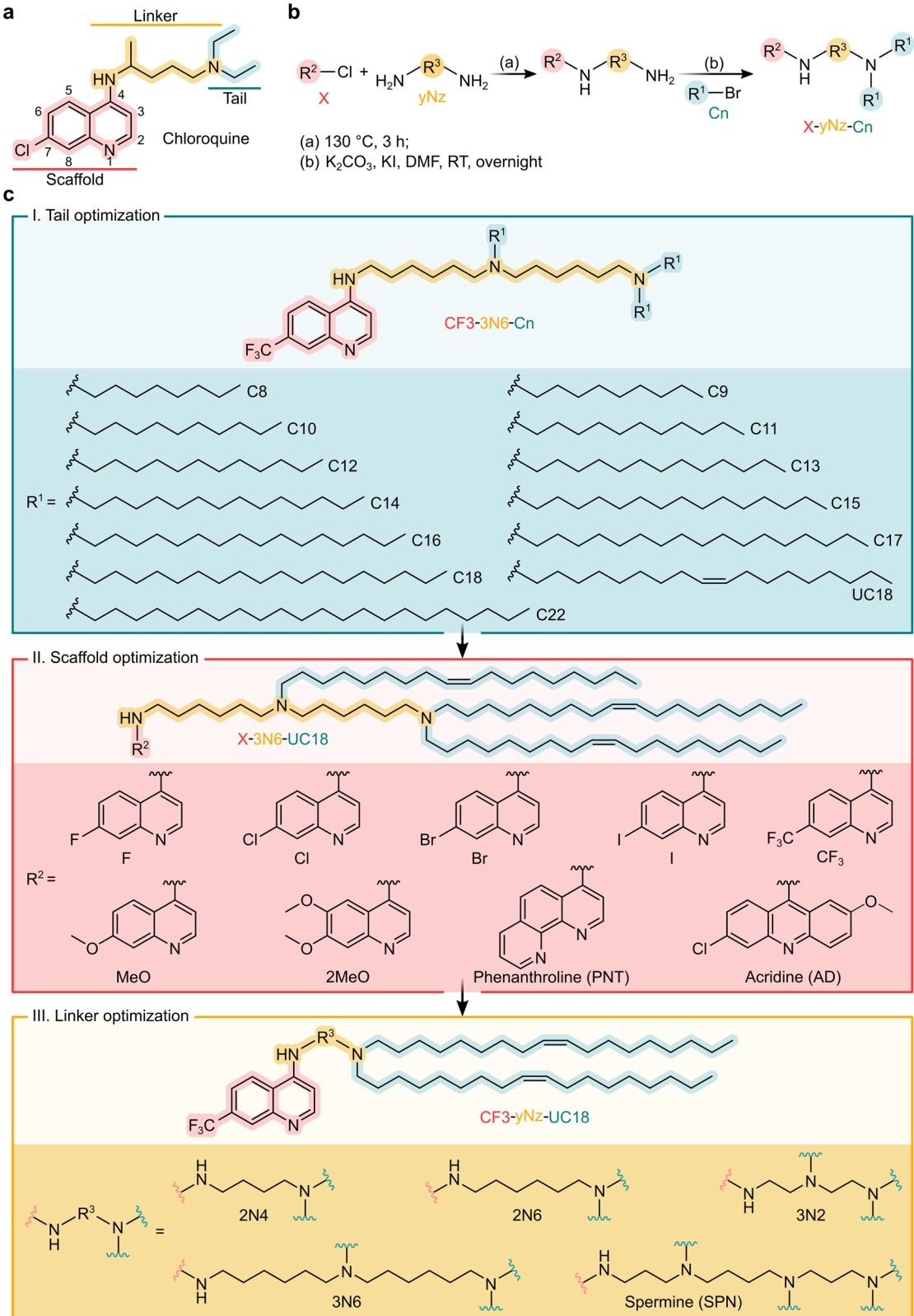

**Fig. 2 | Modular design and synthesis of CIIs. a** Chloroquine was divided into three segments based on its chemical structure and those of designed CIIs. **b** Synthetic route of CIIs. RT, room temperature. **c** Modular design of CIIs towards improved mRNA delivery activity through three rounds of iterative optimization.

the amine as a tail, and an ionizable diamine linker connecting the active group and alkyl tails (Fig. 2a). A chemically diverse library of 45 CIIs built from combinations of three building blocks was synthesized by attaching the ionizable linker to the 4-position of the 7-substituted quinoline ring and subsequently introducing the alkyl tails via

nucleophilic substitution with alkyl bromides (Fig. 2b). Chemical structures of all compounds are shown in Supplementary Fig. 1. IUPAC names determined using ChemDraw and relevant chemical information are outlined in Supplementary Table 1. For simplicity, these compounds were named X-yNz-Cn (X = the substituent group on the

quinoline ring or the abbreviation of N-heterocyclic ring similarly to the quinoline ring; y = number of nitrogen atoms in the linker; N = nitrogen atom; z = number of carbon atoms between adjacent nitrogen atoms in the linker; n = number of carbon atoms in the alkyl chain) according to their building blocks (Fig. 2b, c).

In the first set of experiments, we sought to create a small library of Clls (from CF3-3N6-C8 to CF3-3N6-C22) via altering the length of alkyl tails (from C8 to C22) while keeping the other two building blocks (the CF3-substituted quinoline ring and 3N6 linker) constant (Fig. 2c). The mRNA delivery performance of these lipids was initially evaluated by encapsulating firefly luciferase mRNA (FLuc mRNA) into delivery vehicles consisting of Clls, helper lipids (DOPE), cholesterol (Chol), and PEGylated lipids (DMG-PEG2k), according to the previously reported ethanol injection method[29]. By monitoring mRNA-encoded FLuc expression in 293T cells, we found that CF3-3N6-C12 and CF3-3N6-C17 showed significantly higher mRNA delivery efficiency than other Clls (Fig. 3a).

After lipid screening, Central Composite Design (CCD), a powerful methodology, was applied to evaluate interactive effects of the four parameters (factors) studied including Cll:mRNA (wt:wt), DOPE:Cll (mol:mol), Chol:Cll (mol:mol), and DMG-PEG2k:Cll (mol:mol). Thirty formulations (runs) with six replicates of the central points were obtained by assigning five ratios (levels) to each factor (Supplementary Fig. 5a and Supplementary Table 2). The result from the first round of four-factor, five-level CCD performed using CF3-3N6-C12 indicated that the optimum formulation in this round was formulation 8 (1st-F8, Fig. 3b, Supplementary Fig. 5a, and Supplementary Table 2) and the weight ratio between Cll and mRNA and molar ratio between Chol and Cll had greater impacts on delivery activity than the other two factors tested (Fig. 3c and Supplementary Fig. 6a). We then conducted the second round of CCD optimization (Supplementary Fig. 5b and Supplementary Table 3) by adjusting levels of the factors based on the first-round result. Under this circumstance, formulation F5 (2nd-F5) with further elevation in mRNA delivery efficiency was identified as the optimum formulation in this round (Fig. 3b, Supplementary Fig. 5b, and Supplementary Table 3), and similar impact trend was observed for four factors (Fig. 3c and Supplementary Fig. 6b). Subsequently, the third round of CCD (Supplementary Fig. 5c and Supplementary Table 4) was carried out using CF3-3N6-C17 as the ionizable lipid and the composition ratios from the second round of CCD as the levels. In this round, one formulation (Formulation 14, 3rd-F14) with the weight ratio of 4:1 for CF3-3N6-C17:mRNA and molar ratio of 15.3:38.4:46.1:0.2 for CF3-3N6-C17:DOPE:Chol:DMG-PEG2k outperformed all other formulations (Fig. 3b, c, and Supplementary Fig. 6c). Given the significant changes in composition ratios between the original formulation and 3rd-F14, all lipids were subjected to re-screen using the composition ratios of 3rd-F14. Of these, a Cll bearing an unsaturated cis-double bond in each C18 tail (CF3-3N6-UC18) showed the most potent delivery activity (Fig. 3d). To further reduce the particle size, formulation compositions with a higher molar percentage of DMG-PEG2k (0.8%) were used for the following module optimization (Supplementary Fig. 7a).

We proceeded to the second set of experiments for quinoline scaffold optimization. Additional two dozen of Clls were synthesized by combining the 3N6 linker with three types of hydrophobic tails (C12, C17, and UC18) and eight types of quinoline ring-based scaffolds (Fig. 2c). Compared with 7-trifluoromethyl substitution (CF3), 6,7-dimethoxyl substitution (2MeO) in the quinoline ring led to a slight increase in mRNA delivery activity for C12-series lipids, and all substituents except the N-heterocyclic ring improved delivery performance for C17-series lipids (Fig. 3e). Yet, in the case of UC18-series lipids, all other substitutions in the quinoline ring reversed this effect (Fig. 3e). Comprehensive analysis of categorical data indicated that UC18-series lipids and lipids containing the CF3-substituted quinoline scaffold performed best (Supplementary Fig. 7b, c). In the next step

towards linker optimization, four polyamine linkers with varied numbers of ionizable nitrogen atoms and/or decreased distances between adjacent nitrogen atoms were used to create the final eight Clls (Fig. 2c). One Cll named CF3-2N6-UC18 displayed the highest mRNA delivery activity relative to all other linker-series lipids (Fig. 3e and Supplementary Fig. 7d). A cross-correlation analysis of the relationship among the relative luminescence intensity (RLI), particle size, and zeta potential of mRNA formulations, as well as the topological polar surface area (tPSA) and calculated octanol−water partition coefficient (clogP) of Clls revealed that there was only a weak positive correlation between the particle size and zeta potential (r = 0.41, $p$ = 0.013, Supplementary Fig. 7e). This result suggested that the physicochemical parameters examined in this study were not ideal indices to predict formulations' mRNA delivery efficiency. Thus far, the lead Cll, CF3-2N6-UC18, and its optimal formulation, CF3-2N6-UC18:mRNA (wt:wt) = 4:1 and CF3-2N6-UC18:DOPE:Chol:DMG-PEG2k (mol:mol:mol:mol) = 15.2: 38.2:45.8:0.8, was identified. We referred to this delivery system hereinafter as endosomolytic chloroquine-like optimized lipid nanoparticles (ecoLNPs).

## Physicochemical and biological characterization of ecoLNPs

The top-performing formulation, ecoLNPs, prepared by manual pipetting possessed high encapsulation efficiency (EE, 98%) and exhibited spherical morphology with a z-average hydrodynamic diameter ($D_H$) of 200 nm, a polydispersity index (PDI) of 0.2, and an average zeta potential (ζ) of 25.8 mV (Fig. 4a and Supplementary Fig. 8a). To provide a direct measure of the stability of ecoLNPs, we replaced the above FLuc mRNA payload with eGFP mRNA and monitored the activity of eGFP mRNA-loaded ecoLNPs stored at different temperatures. They were found to be stable for at least 8 days under non-frozen conditions, as evidenced by the negligible changes in the particle size (Supplementary Fig. 8b) and robust GFP expression in 293T cells treated with formulations stored at 4 °C during measurements (Fig. 4b). However, a further increase in storage temperature posed a negative effect on formulations' delivery efficiency, albeit that no significant changes with time in the particle size occurred (Supplementary Fig. 8b, c). These results suggested that non-frozen cold storage was required to preserve formulations' stability and activity.

Considering that mRNA molecules undergo serum- and RNase-induced degradation in vivo before reaching the desired target tissues and cells, we tested the capacity of ecoLNP-encapsulated mRNA to resist degradation by incubation with 10% fetal bovine serum or an ultra-high concentration of RNase A (50 μg ml$^{-1}$). Free mRNA was degraded within 10 min when exposed to serum or even a low concentration of RNase A (0.5 μg ml$^{-1}$), whereas CF3-2N6-UC18 ecoLNP-encapsulated mRNA was resistant to both serum and RNase A over 6 h (Fig. 4c).

The pH-dependent in vitro hemolysis testing revealed that ecoLNPs did not cause visible hemolytic effect on erythrocytes at normal physiological pH (~7.4), even under higher concentrations (2.5× and 6.25× relative to the formulations used in the in vitro assays) (Fig. 4d). This property makes ecoLNPs good candidates as biocompatible vehicles for mRNA delivery. By contrast, the hemolysis rate at endo-lysosomal pH (~5.4) was substantially increased with increasing ecoLNPs concentrations and reached 43.9% at the highest concentration tested (6.25×, Fig. 4d). This result provided initial evidence that ecoLNPs might be pH-sensitive endosome-disruptive vehicles as the hemolytic activity of a vehicle at the pH range of endosomes positively correlates with its ability to escape from endosomes.

After characterizing the physicochemical properties of the top-performing mRNA delivery system, we turned our attention to its detailed biological activity. When loaded with FLuc mRNA, ecoLNPs were found to be 18.9- and 4.4-fold more potent in delivery performance than commercially available transfection reagents Lipo2k and LipoMMAX, respectively (Supplementary Fig. 8d). In addition to FLuc

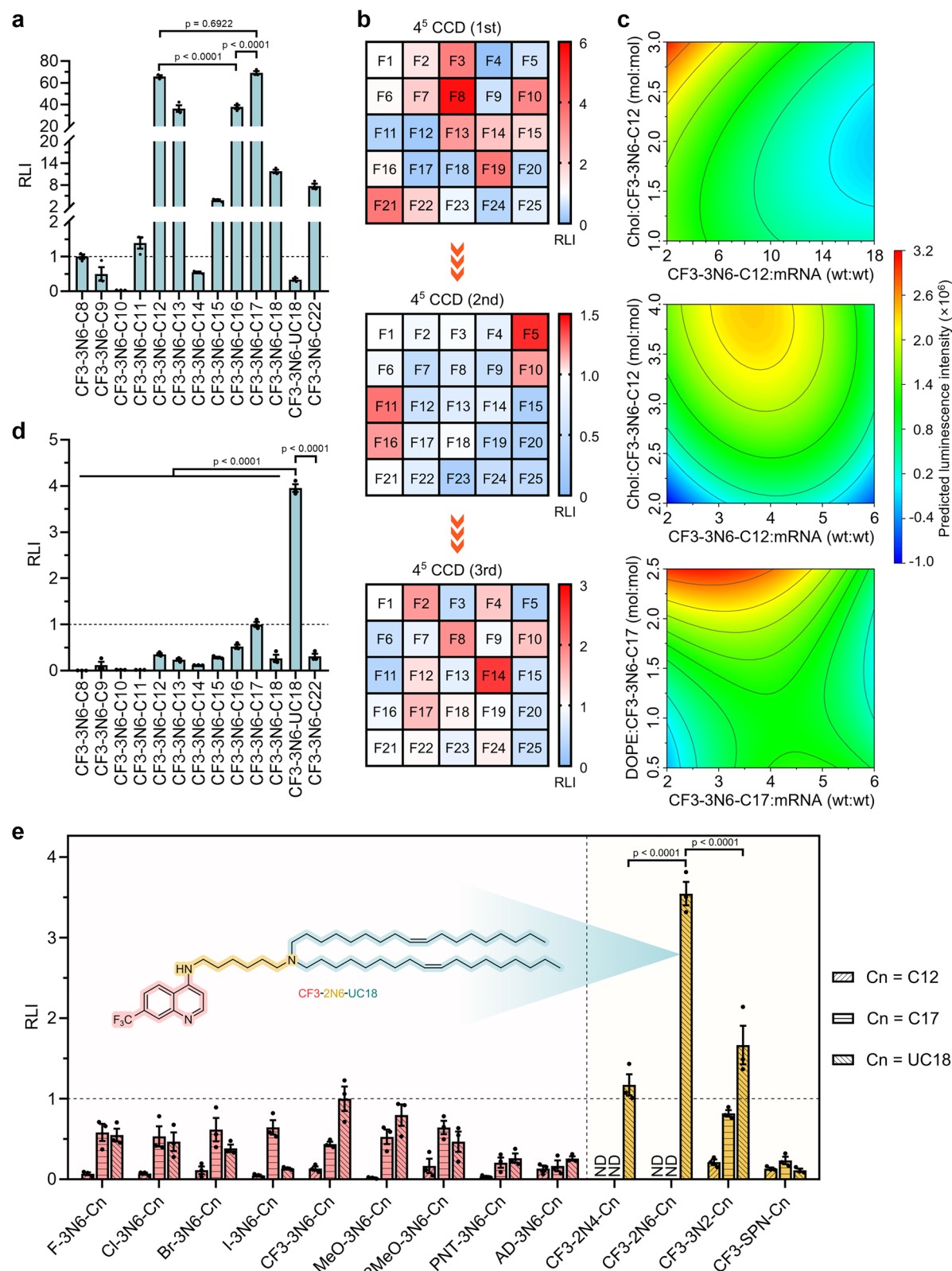

mRNA, enhanced green fluorescent protein mRNA (eGFP mRNA) was also used as a reporter for the subsequent studies. There is a dose and time response for ecoLNPs-mediated delivery activity within 24 h and with doses ≤200 ng per well in 293T cells, whereas a further increase in dose or exposure time did not result in a higher fluorescence intensity (Fig. 4e). The percentage of eGFP-positive 293T cells following ecoLNP treatment was higher than 90% even under a low mRNA dose (50 ng) or short incubation time (6 h) (Supplementary Fig. 8e), and 3- and 2.2-fold higher at a mRNA dose of 200 ng than that of Lipo2k and LipoMMAX after 24 h of treatment (Fig. 4f and Supplementary Fig. 8f). The trend was more pronounced for hard-to-transfect human monocytic cells (Fig. 4f and Supplementary Fig. 8f). By using the other two

**Fig. 3 | Optimization of mRNA formulations using CCD. a** Screening of CIIs bearing varied lipid tails for mRNA delivery in 293T cells. FLuc mRNA formulated with CIIs and auxiliary lipids was used for tail screening. The relative luminescence intensity (RLI) was obtained by normalizing the luminescence intensity of each group to that of the CF3-3N6-C8 group ($n = 3$ biological replicates, one-way ANOVA with Tukey's multiple comparison test). **b** Three rounds of $4^5$ CCD applied to optimization of formulation compositions. A total of 25 formulations (F1–F25) (30 formulations with six replicates for the central point) were performed in each round. The luminescence intensity of each group was normalized to that of the internal central points (formulation 1, F1). The first two rounds of CCD were applied to CF3-3N6-C12 (top and middle panels), and the third one to CF3-3N6-C17 (bottom panel). **c** Effect of five levels from the top two factors on the delivery activity of mRNA formulations across three rounds of $4^5$ CCD. Trends in intensity changes were predicted by the CCD program. The levels (ratios) of the other two factors (DOPE:CII and DMG-PEG2k:CII) were fixed as those of the optimum formulation measured. **d** Re-screening of CIIs bearing varied lipid tails for mRNA delivery using the formulation composition of 3rd-F14. The luminescence intensity of each group was normalized to that of the CF3-3N6-C17 group ($n = 3$ biological replicates, one-way ANOVA with Tukey's multiple comparison test). **e** Screening of CIIs bearing varied scaffolds or linkers for mRNA delivery. The luminescence intensity of each formulation was normalized to that of the CF3-3N6-UC18 group ($n = 3$ biological replicates, two-way ANOVA with Tukey's multiple comparison test). ND, not determined. The chemical structure of the lead lipid CF3-2N6-UC18 was shown. For all relevant panels, data are presented as mean ± SEM. Source data are provided as a Source Data file.

mRNA reporters, mCherry mRNA and galactosidase mRNA (β-gal mRNA), we further demonstrated the robust mRNA delivery activity mediated by ecoLNPs (Fig. 4g, h). Aside from mRNA payloads, we also tested the delivery performance of ecoLNPs towards other nucleic acid molecules including siRNA and plasmid DNA. For siRNA delivery, a greater than 85% knockdown of luciferase expression was observed for cells stably expressing the luciferase reporter even at the siRNA concentration as low as 1 nM after 24 h of treatment with ecoLNPs-encapsulated siRNA targeting the luciferase reporter (Supplementary Fig. 8g). Efficient gene delivery was also achieved when plasmid DNA was formulated into ecoLNPs (Supplementary Fig. 8h). Lastly, we attempted to deliver reporter mRNA to 3D cells mimicking tissue microarchitecture, which was considered hard to transfect due to limited penetration[30]. Unexpectedly, we observed that delivery of eGFP mRNA, mCherry mRNA (Fig. 4i), or co-delivery their combinations (Fig. 4j) to 3D cells led to successful mRNA translation throughout the entire 3D structure.

## Endosomolytic activity mediated by ecoLNPs

Before probing the endosomolytic action of ecoLNPs, we incorporated mRNA payloads labeled with the Cy5 fluorescent probes into ecoLNPs as the first step toward visualization. We subsequently investigated their internalization process using both confocal microscopy and flow cytometry and found that ecoLNPs were almost completely internalized by cells within a few hours (Supplementary Fig. 9a, b). A brief preincubation of cells with specific inhibitors of endocytic pathways including chlorpromazine (CPZ, blocking clathrin-mediated endocytosis), filipin (blocking caveolae-mediated endocytosis), and 5-(N-ethyl-N-isopropyl)-amiloride (EIPA, blocking macropinocytosis) caused distinct extents of reductions in the mean Cy5 fluorescence intensity (MFI) in the rank order EIPA > filipin > CPZ (Fig. 5a and Supplementary Fig. 9a). The inhibitory effects demonstrated that cellular entry of ecoLNPs involved multiple energy-dependent endocytic pathways, especially macropinocytosis.

Aside from the above inhibitors, Bafilomycin A1 (Baf), a specific V-type H⁺-ATPase inhibitor, was utilized to explore the effects of proton pumps on the cellular uptake of ecoLNPs. The dramatic decrease in the MFI in the presence of two different concentrations of Baf (Fig. 5a) indicated that energy-dependent proton pumps are the dominant driving force of ecoLNPs entry into cells. The result was in agreement with the previous observation[31] and could plausibly be attributed to the blockade of ecoLNPs transport from endosomes to lysosomes or escape from endosomes, and subsequent disturbance in endocytosis and intracellular trafficking[32]. In addition, the low internalization rate of ecoLNPs induced by low-temperature incubation further reflected that transport of ecoLNPs was an energy-dependent process.

As CF3-2N6-UC18 serves as a weak Brønsted base capable of accepting a proton from acidic conditions and its internalization as well as endosomal escape can be blocked by the proton pump inhibitor Baf, it is reasonable to infer that the endosomolytic action of ecoLNPs is relevant to the proton sponge effect, a hypothesis to explain endosomal escape of vehicles with the large buffering capacities via buffering endo-lysosomes, increasing the influx of protons, and causing the rupture of endo-lysosomal membranes[33]. To seek further evidence on the proton sponge effect, we calculated the buffering capacity of CF3-2N6-UC18 from its titration curve (Supplementary Fig. 10a, b). The high buffering capacity of CF3-2N6-UC18 at endo-lysosomal pH range indicated that it possesses the potential to serve as a proton sponge to absorb a substantial amount of protons. As an alternative approach to investigate the proton sponge effect is to measure the endo-lysosomal pH of living cells after treatment, in the following experiment, we incubated HeLa cells with ecoLNPs and measured the endo-lysosomal pH at two specific time points using FITC-Dextran and Alexa Fluor 647-Dextran[34]. The colocalized yellow fluorescence signals detected in the merged confocal fluorescence images revealed that incubation of cells with ecoLNPs led to a significant increase in the endo-lysosomal pH when compared with PBS-treated cells (Supplementary Fig. 10c, d). This observation along with the similar results from other reagents that function as proton sponges[35–37] again provided direct evidence to support the proton sponge effect mediated by ecoLNPs.

After establishing the internalization mechanism of ecoLNPs, we performed subcellular colocalization assays involved in the intracellular trafficking with endo-lysosomal fluorescent acidotropic probes LysoTracker green. In this stage, PNT-2N6-UC18, a CF3-2N6-UC18 analog bearing a phenanthroline ring instead of a quinoline ring, was synthesized as a control lipid (Fig. 5b). LNPs formulated with PNT-2N6-UC18 were efficiently taken up by the cells but showed poor mRNA translation efficiency (Supplementary Fig. 9c). Using confocal fluorescence microscopy, we found that the Cy5 fluorescence signals were largely colocalized with endo-lysosomes in PNT-2N6-UC18 LNP-treated cells, but evenly distributed throughout the cytoplasm with a minor overlap with LysoTracker green signals in those treated with ecoLNPs (Fig. 5c, d). The differences in colocalization between PNT-2N6-UC18 LNPs and ecoLNPs suggested that the quinoline ring should play a crucial role in endosomal escape.

We next carried out another fluorescence assay using membrane-impermeable fluorescent dyes calcein as a reporter to further verify the endosomal escape capacity of ecoLNPs. Endocytosed calcein shows punctate fluorescence signals as it is prone to be entrapped within the endo-lysosomal compartments but brightens throughout the cytoplasm after leaking from endo-lysosomes[38]. Compared with the punctate distribution patterns of calcein occurring in cells treated with free mRNA or PNT-2N6-UC18 LNPs, the diffuse calcein fluorescence signals from the cells exposed to ecoLNPs clearly demonstrated that massive calcein leakages from endosomes into the cytoplasm were boosted by the pH-sensitive endosome-disruptive activity induced by ecoLNPs (Fig. 5e, f). This result again demonstrated the importance of the quinoline ring in endosomal escape of engulfed ecoLNPs.

SapB is a membrane-perturbing and lipid-binding lysosomal protein that involved in lipid transport and is shown to bind

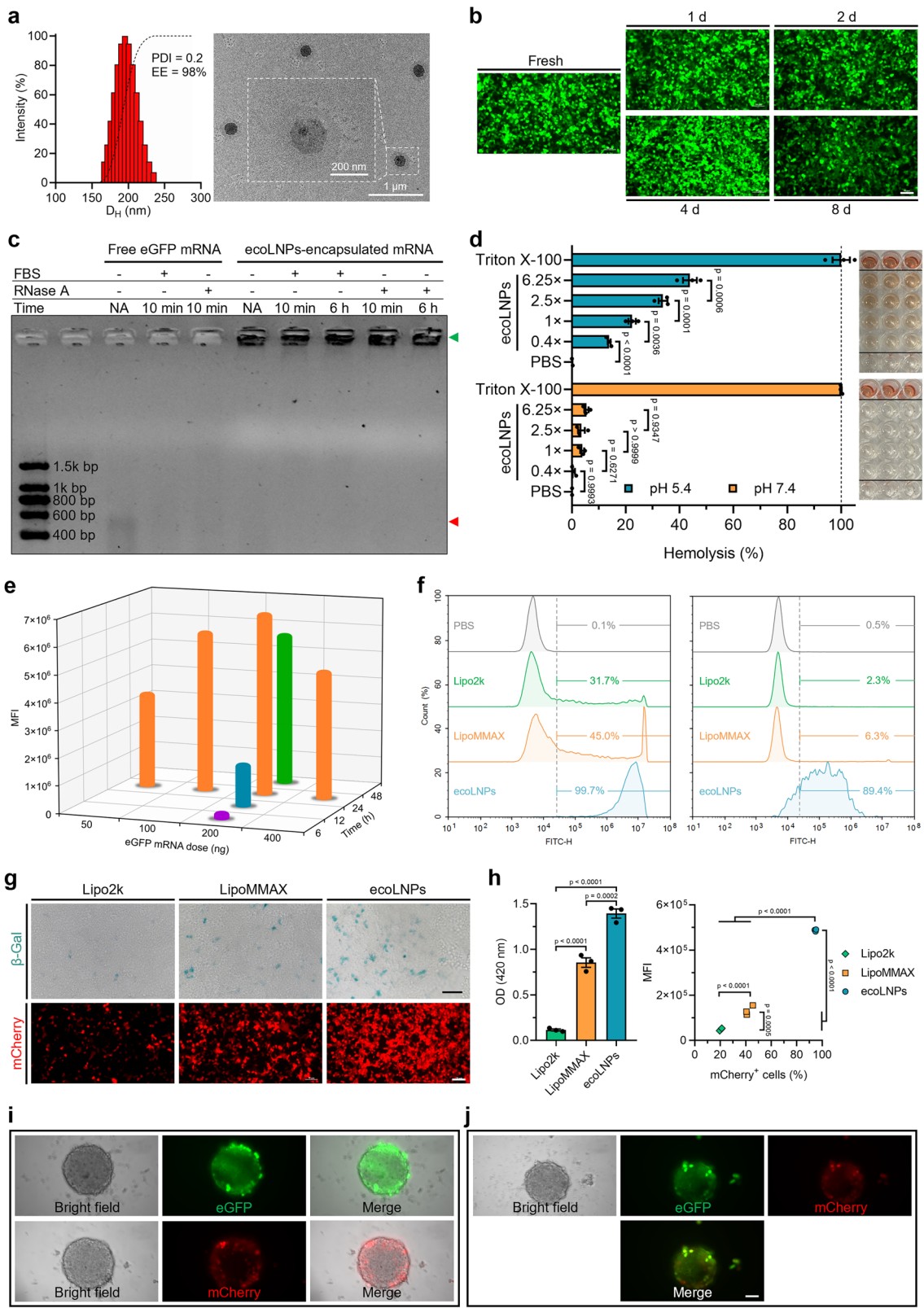

chloroquine in the dimeric form refs. 39–41. To gain additional insights into the differences in endosomal escape capacity between CF3-2N6-UC18 and PNT-2N6-UC18, we performed the molecular docking of each lipid to the binding pocket of lysosomal sapB in silico based on the crystal structure of sapB with bound chloroquine[40]. When docking into the dimeric interface of sapB, CF3-2N6-UC18 displays a similar

conformation to the bound chloroquine (Fig. 5g). That is, the quinoline ring and linker of CF3-2N6-UC18 fits well with the hydrophobic pocket formed by the residues M61, M65, L73, E35, and R38, and the long tail is oriented to the giant cavity (Fig. 5g). Nevertheless, replacement of the quinoline scaffold with phenanthroline (PNT) leads to a highly constrained conformation for PNT-2N6-UC18 (the binding energy is

**Fig. 4 | Characterization of physicochemical properties and mRNA delivery performance of ecoLNPs. a** The z-average hydrodynamic size distribution (left panel) and representative transmission electron microscopy images (right panel) of ecoLNPs. **b** In vitro delivery performance of ecoLNPs during storage at 4 °C. ecoLNPs containing 200 ng of eGFP mRNA were stored at 4 °C for different storage times and their in vitro delivery performance was estimated by the intensity of green fluorescence originating from ecoLNPs-treated 293T cells. Scale bar, 50 μm. **c** Susceptibility of ecoLNPs-encapsulated eGFP mRNA to serum or RNase A. Free eGFP mRNA was used as a control group. The green and red triangles indicate the location of bands of ecoLNPs-encapsulated and free mRNA, respectively. **d** Hemolytic activity induced by different concentrations of ecoLNPs at pH of 5.4 and 7.4 ($n = 3$ biological replicates, two-way ANOVA with Tukey's multiple comparison test). The concentration of ecoLNPs used for assessment of in vitro delivery activity was defined as 1×. **e** Dose-response and time-course analyses of 293T cells exposure to eGFP mRNA-loaded ecoLNPs. MFI, mean fluorescence intensity. **f** A comparison of eGFP mRNA delivery efficiency of ecoLNPs with Lipofectamine 2000 (Lipo2k) and Lipofectamine MessengerMAX (LipoMMAX) in 293T (left panel) and THP-1 cells (right panel) by flow cytometry. PBS-treated cells were used to differentiate positive and negative events. **g** A comparison of β-gal mRNA delivery efficiency of ecoLNPs with Lipo2k and LipoMMAX in 293T cells by in situ β-galactosidase staining (top panel; scale bar, 100 μm) and a comparison of mCherry mRNA delivery efficiency by fluorescence imaging (bottom panel; scale bar, 50 μm). **h** Quantitative analysis of both β-gal mRNA and mCherry mRNA delivery efficiency using a colorimetric method (left panel) and flow cytometry (right panel), respectively ($n = 3$ biological replicates, one-way ANOVA with Tukey's multiple comparison test). MFI, mean fluorescence intensity. Fluorescent images of 3D cells treated with eGFP mRNA- (**i**, top panel), mCherry mRNA- (**i**, bottom panel), or dual mRNA- (**j**) loaded ecoLNPs. Scale bar, 50 μm. For all relevant panels, data are presented as mean ± SEM. Source data are provided as a Source Data file.

−4.67 kcal mol⁻¹ for PNT-2N6-UC18 and −5.24 kcal mol⁻¹ for CF3-2N6-UC18) (Fig. 5g). The differences in the binding pattern and energy between CF3-2N6-UC18 and PNT-2N6-UC18 suggested that the interaction between CF3-2N6-UC18 and sapB may facilitate CF3-2N6-UC18 exchange across endo-lysosomal membranes, thereby promoting endo-lysosomal fusion and ultimate escape.

Taken together, the computer-assisted molecular docking along with the pH-dependent in vitro hemolytic activity and the endosomal escape assay supported our hypothesis that a triple combination of a quinoline scaffold, ionizable nitrogen atoms and hydrophobic lipids tail should yield a chloroquine-like lipid with pH-sensitive endosome-disruptive activity for mRNA delivery.

## ecoLNPs as a robust platform for mRNA delivery in vivo

The unique pH-responsive endosomolytic behavior of ecoLNPs encouraged us to continue exploring their mRNA delivery efficacy in vivo. Since the pattern of LNP bio-distribution can be markedly affected by the routes of administration, we first monitored the in vivo biodistribution and accumulation of ecoLNPs containing Cy5-labeled mRNA in mice following three types of common routes including intravenous (IV), subcutaneous (SC), and intramuscular (IM) injection. Four hours after administration, bright fluorescence signals were observed at the injection sites following SC and IM injection, whereas the majority of signals were detected in the liver and spleen for IV injection (Supplementary Fig. 11a, b). A further analysis of their histological sections also confirmed their bio-distribution profiles (Supplementary Fig. 11c, d).

To determine whether mRNA delivery to the specific tissues can be translated into corresponding proteins, we replaced the Cy5-labeled mRNA payloads with FLuc mRNA and checked their bioluminescence signals in mice under the same conditions. In accordance with the above observations, we found that mRNA-encoded FLuc were expressed at very high levels exclusively at the SC and IM injection sites, while typically located in the mice liver and spleen under IV injection (Fig. 6a). Of note, ex vivo bioluminescence imaging indicated that mRNA delivered by ecoLNPs via IM injection were prominently expressed in lymph nodes rather than other tissues examined (Fig. 6a). The lymph node tropism of ecoLNPs should make this vehicle an attractive candidate for immunotherapy. In addition to the common administration routes, we also assessed other routes including subconjunctival (SCJ) and subretinal (SR) injections as well as intravesical (IVS) instillation as an extension. Tissue-specific expression of mRNA-encoded proteins was observed for mice receiving ecoLNPs via all three local administration routes (Fig. 6a and Supplementary Fig. 12a). Collectively, ecoLNPs were proven to be versatile as a mRNA delivery platform.

Apart from multiple administration routes, we also evaluated the range of dosage required for systemic mRNA delivery in mice. A dose-dependent mRNA delivery activity mediated by ecoLNPs was observed with a single mRNA dose ranging from 0.03 mg kg⁻¹ to 0.5 mg kg⁻¹

(Supplementary Fig. 12b, c). Treatment with ecoLNPs at 0.13 mg kg⁻¹ or higher doses achieved superior mRNA delivery potency (Supplementary Fig. 12b, c). Moreover, FLuc activity was also detectable even at the lowest dose tested (0.03 mg kg⁻¹) (Supplementary Fig. 12b, c). In addition, we investigated the expression kinetics of ecoLNPs-encapsulated mRNA delivered through the IV route. A single intravenous injection of FLuc mRNA-loaded ecoLNPs at a mRNA dose of 0.5 mg kg⁻¹ resulted in robust FLuc expression in the liver and spleen 2 h after injection (Supplementary Fig. 12b, d). The bioluminescence signals remained at a high level for at least 8 h and could still be detected in the liver and spleen before day 3 (Supplementary Fig. 12b, d, e). In vivo safety evaluation of ecoLNPs in mice indicated that a single IV injection did not cause significant alterations in liver and kidney function as well as main serum biochemistry parameters (Supplementary Table 5). Furthermore, no obvious pathological changes occurred in major tissues following IV, SC, and IM injection (Supplementary Fig. 13).

We next compared the delivery efficiency of ecoLNPs with that of clinically approved SM-102 LNPs which was used as an intramuscularly injected vehicle for the Moderna's COVID-19 mRNA vaccine. FLuc mRNA used at the same dosage as ecoLNPs (0.5 mg kg⁻¹) was formulated strictly into SM-102 LNPs according to the reported nano-formulation recipe[42,43] and administrated into the mice through IM injection. As shown in Fig. 6b–d, ecoLNPs displayed similar delivery potency to SM-102 LNPs but higher tropism toward lymph nodes (90.2% vs 58.1%). Since most mRNA vaccines require ultra-cold temperatures for storage and transportation, we subsequently investigated the stability of ecoLNPs stored at 4 °C and found that ecoLNPs were able to maintain their in vivo delivery potency for at least eight days (Supplementary Fig. 14a, b). The stability was in line with that estimated by monitoring the changes of the particle size and in vitro delivery performance (Fig. 4b and Supplementary Fig. 8b).

Beyond mRNA reporters, we also evaluated the utility of ecoLNPs for delivery of functional mRNA. A single IM injection of ecoLNPs-encapsulated mRNA encoding Cre recombinase (Cre mRNA) at a mRNA dose of 0.5 mg kg⁻¹ into Ai9 Cre-tdTomato reporter mice[44] (Fig. 6e) led to a 4.7-fold increase in tdTomato fluorescence signals at the injection sites compared to untreated legs (Fig. 6g and Supplementary Fig. 14c). The activation of tdTomato reporter was also observed in tissue sections (injected muscles and lymph nodes, Fig. 6i) and validated by gel electrophoresis (Fig. 6j). These results implied that the delivered Cre mRNA at the target tissues was successfully translated into Cre proteins and participated in excision of the stop cassette flanked by the loxP sites. Finally, we attempted to survey the genome editing activity of ecoLNPs via co-delivery of CRISPR-Cas9 mRNA and gRNA (Fig. 6f). Since gRNA simultaneously targeting the three repeats of the loxP-flanked stop cassette theoretically generates multiple types of edits and only double repeat deletion can activate tdTomato expression[45], it is considered technically more challenging

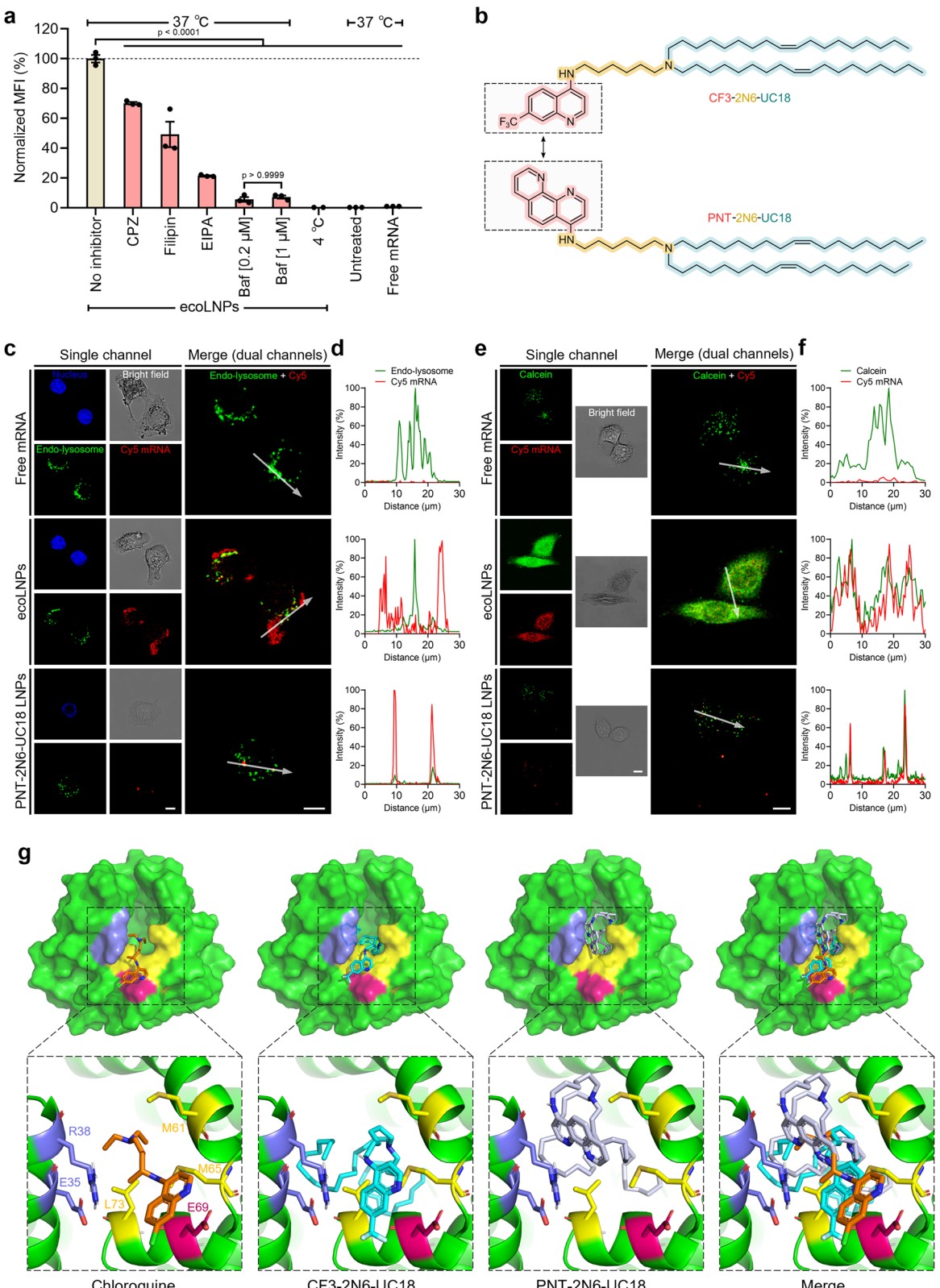

to detect tdTomato fluorescence signals than the Cre-loxP system (Fig. 6f). Even so, the activation of tdTomato expression resulted from site-specific genome editing was achieved through inducing double strand breaks on three identical target sites of the stop cassette and subsequent non-homologous repair (Fig. 6h, i, and Supplementary Fig. 14d). Ultimately, eleven types of edits in the injection sites and

their genome editing frequencies were confirmed by third-generation sequencing (Supplementary Fig. 15a, b).

Taken together, these data indicated that ecoLNPs were capable of delivering mRNA in vivo potently regardless of routes of administration, without inducing common adverse effects. As biodegradable lipids tend to degrade more rapidly than those without degradable bonds[13,46,47],

**Fig. 5 | Endosomal escape of ecoLNPs. a** Effects of inhibitors and temperature on the cellular uptake of ecoLNPs. MFI of each group was determined by flow cytometry and normalized to that of the "no inhibitor" group. Data are presented as mean ± SEM ($n$ = 2 biological replicates for 4 °C, $n$ = 3 biological replicates for the remaining groups, one-way ANOVA with Tukey's multiple comparison test). **b** A comparison of chemical structures between CF3-2N6-UC18 and PNT-2N6-UC18. Representative confocal microscopy images of HeLa cells treated with Cy5 mRNA (red)-loaded ecoLNPs for tracking ecoLNP-mediated endosomal escape via co-localization analysis (**c**) and the calcein leakage assay (**e**). Over a dozen of images were taken for each group and the most representative image was presented. Endo-lysosomal tracker (green) and Hoechst (blue) were used to stain endo-lysosomes and nucleus (**c**). Calcein (green) leakage was used to evaluate the endosomal escape capacity of co-incubated molecules (**e**). Free Cy5 mRNA and Cy5 mRNA-loaded PNT-2N6-UC18 LNPs served as control groups. The red fluorescence signals in the PNT-2N6-UC18 LNPs-treated group appeared weak at high magnification (63× oil immersion objective lens), largely owing to their punctate distribution. Scale bar, 10 μm. Quantitative analysis of fluorescent colocalization of Cy5 and endo-lysosomal tracker fluorescence (**d**) in (**c**), or calcein fluorescence (**f**) in (**e**) across white arrows (30 μm). The maximum fluorescence intensity was defined as 100% in each panel. **g** Molecular docking of CF3-2N6-UC18 (sticks in cyan) or PNT-2N6-UC18 (sticks in gray) to the sapB dimer binding pocket (surface representation in green). Chloroquine (sticks in oranges) in complex with sapB was used as a reference. The hydrophobic residues (M61, M65, and L73 from one monomer; E35 and R38 from the second monomer) were labeled in yellow and blue, respectively. The residue E69 involved in the potential hydrogen-bonding interaction with chloroquine was labeled in magenta. Inset, the interaction details of chloroquine or lipids with sapB. Single-letter abbreviations for amino acid residues are as follows: E, Glu; L, Leu; M, Met; and R, Arg. Source data are provided as a Source Data file.

future work will focus on designing biodegradable counterparts and comparing their efficacy and safety to lipids developed in this study.

## Discussion

Here, we designed and synthesized a series of chloroquine-like lipids with signature quinoline scaffolds, ionizable linkers, and lipid tails. According to the present structure activity relationship, we proposed several general criteria for designing highly active Clls. (1) Subtle variations in the length of carbon chain influence delivery potency of Clls. (2) Disruption of the quinoline scaffold by losing their signature quinoline rings, such as PNT and AD series, impair Clls' activity. (3) A strong preference for substitution at the 7-position of the quinoline ring is the trifluoromethyl (CF3) group. (4) Introduction of excess ionizable nitrogen atoms linked with lipid tails may negatively impact delivery. (5) Three modules (quinoline scaffolds, ionizable linkers, and lipid tails) of Clls synergistically affect their delivery performance. Thus, a triple combination will be required to produce Clls with more robust mRNA delivery efficiency.

Formulation optimization through four-factor, five-level CCD of response surface methodology indicated that the weight ratio between Cll and mRNA and molar ratio between Cll and Chol had a great influence on mRNA delivery efficiency. The lead lipid CF3-2N6-UC18 comprised of a 7-trifluoromethyl substituted quinoline scaffold, a hexamethylenediamine linker, and two unsaturated oleyl tails was capable of self-assembling into stable ecoLNPs and displayed higher delivery efficiency towards mRNA, siRNA, and plasmid DNA delivery in vitro than three gold standard Lipofectamine transfection reagents.

Mechanistically, the internalization process of ecoLNPs involved multiple energy-dependent endocytosis pathways and the V-ATPase proton pump. The high delivery efficiency of ecoLNPs could be attributed to their high endosomal escape abilities, which appeared to be relevant to the proton sponge effect and sapB-promoted endosomal membrane fusion. In comparison with the formulation prepared with PNT-2N6-UC18 analog lacking the signature quinoline scaffold, ecoLNPs showed significantly reduced co-localization with endo-lysosomes and enhanced calcein release, suggesting that the endosomal escape heavily relied on its quinoline ring.

EcoLNPs were proven to be safe and versatile mRNA carriers for several routes of administration including systemic IV injection and local SC, IM, SCJ, SR injections and IVS instillation. After IM injection into mice, ecoLNPs showed comparable delivery performance to clinically approved SM-102 LNPs, but improved tropism towards lymph nodes. Furthermore, delivery of ecoLNPs-encapsulated Cre mRNA or CRISPR-Cas9 mRNA combined with gRNA resulted in efficient genome editing in Ai9 transgenic mice, further demonstrating their application potential in mRNA delivery in vivo. The structure-guided modular design and optimization methodology presented here should provide a strategy for designing next-generation mRNA delivery systems.

## Methods

### General procedure for the synthesis of Clls

The general procedure was exemplified by the synthesis of the lead Cll, $N^1$, $N^1$-di((Z)-octadec-9-en-1-yl)-$N^6$-(7-(trifluoromethyl)quinolin-4-yl)hexane-1,6-diamine (CF3-2N6-UC18). In brief, to a round-bottom flask equipped with a magnetic stirring bar was added 4-chloro-7-(trifluoromethyl)quinoline (1.0 equiv) and hexamethylenediamine (3.0 equiv). The reaction mixture was heated to 130 °C for 3 h. After being cooled to room temperature, the reaction products were washed with $H_2O$ (15 ml) and extracted with $CH_2Cl_2$ (10 ml). The organic layer was dried over anhydrous $Na_2SO_4$ and concentrated in vacuo. The obtained intermediate (1.0 equiv) was dissolved in DMF (5 ml) and reacted with oleyl bromide (2.4 equiv), $K_2CO_3$ (2.5 equiv), and KI (0.5 equiv) in a round-bottom flask equipped with a magnetic stirring bar. The reaction mixture was allowed to stir at room temperature for about 20 h. Then, $CH_2Cl_2$ (10 ml) was added, followed by washing twice with $H_2O$ (15 ml ×2). The organic layer was dried over anhydrous $Na_2SO_4$, concentrated in vacuo, and purified by a flash chromatography system (SepaBean machine U200, Santai Technologies) under gradient elution from 100% $CH_2Cl_2$ to $CH_2Cl_2$/MeOH mixture (4:1, v/v). The structure of CF3-2N6-UC18 was confirmed by high resolution mass spectrometry (HR-MS, Bruker, Supplementary Fig. 2) and nuclear magnetic resonance (NMR, Bruker, Supplementary Figs. 3 and 4). HR-MS: m/z calculated for $C_{52}H_{88}F_3N_3$ 811.6930; found (M + H)$^+$ 812.6930. $^1$H NMR (400 MHz, CDCl$_3$) δ: ppm 8.62 (d, $J$ = 5.3 Hz, 1H), 8.27 (s, 1H), 7.87 (d, $J$ = 8.7 Hz, 1H), 7.60 – 7.56 (m, 1H), 6.50 (d, $J$ = 5.3 Hz, 1H), 5.46 – 5.27 (m, 4H), 5.15 (s, 1H), 3.34 (dd, $J$ = 12.3, 6.9 Hz, 2H), 2.42 (dd, $J$ = 14.7, 7.2 Hz, 6H), 2.05 – 1.98 (m, 8H), 1.80 (dt, $J$ = 14.3, 7.2 Hz, 2H), 1.54 – 1.21 (m, 54H), 0.88 (t, $J$ = 6.7 Hz, 6H). $^{13}$C NMR (100 MHz, CDCl$_3$) δ: ppm 152.22, 149.46, 147.64, 130.35, 130.23, 129.89, 129.76, 127.66, 127.62, 120.66, 119.99, 100.00, 54.12, 53.93, 43.23, 32.55, 31.85, 29.71, 29.56, 29.52, 29.46, 29.26, 29.23, 29.08, 28.76, 27.61, 27.26, 27.15, 27.03, 26.90, 26.78, 22.63, 14.06. The other Clls were synthesized with their corresponding starting materials using the same procedure. For X-2Nz-Cn, X-3Nz-Cn, and X-SPN-Cn, the molar feed ratios of intermediate (X-yNz):alkyl bromide (Cn):$K_2CO_3$:KI were 1:2.4:2.5:0.5, 1:3.6:3.7:0.5, and 1:4.8:4.9:0.5, respectively. HR-MS data of all Clls were provided in Supplementary Table 1.

### Preparation of lipid-based formulations for mRNA delivery

LNPs were prepared through the ethanol injection method. Briefly, an ethanol phase containing Cll, DOPE (Avanti), cholesterol (Sangon Biotech), and DMG-PEG2k (TargetMol) at various molar ratios as well as 10% (v/v) citrate buffer (50 mM, pH = 3) was mixed with a mRNA solution diluted in citrate buffer (50 mM, pH = 3) at a volume ratio of 1:1 and various weight ratios by manual pipetting. The mixture was then diluted with equal volume of PBS and left at room temperature for 10 min to form mRNA formulations. For in vivo experiments, higher concentrations of formulations (2.5×) were prepared and subjected to further dialysis against PBS buffer using Slide-A-Lyzer MINI Dialysis

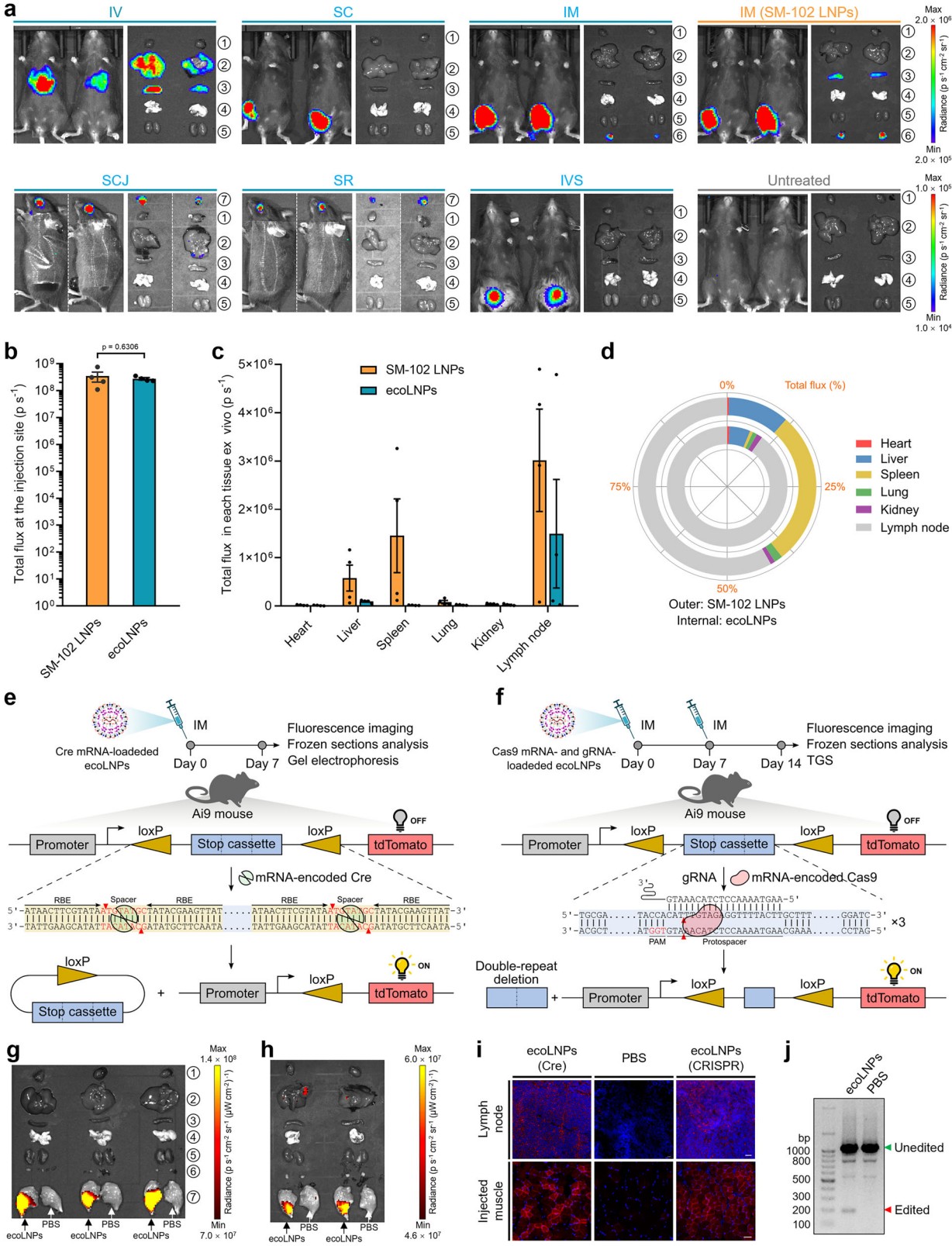

Devices (3.5 K MWCO, Thermo Fisher Scientific). The original formulation consisting of CII, DOPE, Chol, and DMG-PEG2k (22:33.1:44.1:0.8, molar ratio) as well as the mRNA payloads (the weight ratio of CII to mRNA was 10:1) was adopted from TT3 LLNs[29]. The compositions of other formulations obtained through CCD were listed in Supplementary Tables 2–4.

## Characterizations of mRNA formulations

The hydrodynamic size, polydispersity index, and zeta potential of formulations were measured at 25 °C using dynamic light scattering (90Plus PALS, Brookhaven Instruments Corporation) as previously described[29,48]. The encapsulation efficiency of formulations was determined by the RiboGreen assay[29]. The topological polar surface

**Fig. 6 | ecoLNPs-mediated mRNA delivery in vivo. a** Representative in vivo and ex vivo bioluminescent images of C57BL/6 mice 4 h post-injection of ecoLNPs at a FLuc mRNA dose of 0.5 mg kg⁻¹ for IV ($n = 2$), SC ($n = 2$), and IM injection ($n = 4$), 200 ng for SCJ ($n = 3$) and SR ($n = 3$) injection, and 0.25 mg kg⁻¹ for IVS ($n = 3$) instillation. 1, heart; 2, liver; 3, spleen; 4, lung; 5, kidney; 6, lymph node; 7, eye. Quantification of luminescence at the IM injection sites of living mice (**b**) and in ex vivo tissues (**c**) 4 h post-IM injection of FLuc mRNA-loaded ecoLNPs or SM-102 LNPs. Data are presented as mean ± SEM ($n = 4$, unpaired, two-tailed Student's $t$ test). **d** The percentage of luminescence for each ex vivo tissue from (**c**). **e, f** Schematic showing the activation of tdTomato expression in Ai9 transgenic mice by delivering Cre mRNA or co-delivering Cas9 mRNA and gRNA simultaneously targeting the three repeats of the loxP-flanked stop cassette locus. RBE

recombinase binding element, PAM protospacer adjacent motif, TGS third-generation sequencing. **g** Ex vivo fluorescent images of Ai9 mice 7d post-injection of ecoLNPs at a Cre mRNA dose of 0.5 mg kg⁻¹ via intramuscular injection ($n = 3$ legs). Legs injected with PBS were used for comparison. 1, heart; 2, liver; 3, spleen; 4, lung; 5, kidney; 6, lymph node; 7, hind leg. **h** Representative ex vivo fluorescent images of Ai9 mice 14d post-injection of two doses of ecoLNPs at a total RNA (Cas9 mRNA:gRNA = 3:1, wt:wt) dose of 1 mg kg⁻¹ via intramuscular injection ($n = 3$ legs). 1, heart; 2, liver; 3, spleen; 4, lung; 5, kidney; 6, lymph node; 7, hind leg. **i** tdTomato expression (red) in frozen sections of injected muscle and lymph nodes from (**g**) and (**h**). Nuclei were stained by DAPI (blue). Scale bar, 20 μm for lymph node, 50 μm for injected muscle. **j** Verification of Cre mRNA-mediated genome editing by gel electrophoresis. Source data are provided as a Source Data file.

area (tPSA) and calculated octanol−water partition coefficient (clog P) of Clls were calculated using ChemDraw. The stability of ecoLNPs was assessed by continuous monitoring of their hydrodynamic size under various storage times at refrigerator temperature. The morphology of ecoLNPs (5×) was determined by transmission electron microscopy (Hitachi, HT7700)[48].

## Assessment of mRNA delivery efficiency in vitro

Four types of reporter mRNA were used for assessment of formulations' mRNA delivery efficiency in vitro. Transfection reagents, Lipofectamine 2000 (Lipo2k, Thermo Fisher Scientific) and Lipofectamine MessengerMAX (LipoMMAX, Thermo Fisher Scientific), were used as positive controls according to manufacturer's instructions. Human embryonic kidney 293T cells (C6008), human cervical cancer HeLa cells (C6330), and the monocytic leukemia THP-1 cells (C6960) were purchased from Beyotime. 293T cells stably expressing the luciferase reporter (293T-Luc2-tdT, 1101HUM-PUMC000622) were obtained from the National Biomedical Cell-Line Resource. All cells were maintained in ATCC-recommended medium supplemented with 10% FBS (Sigma), 100 U ml⁻¹ penicillin (Beyotime), and 100 μg ml⁻¹ streptomycin (Beyotime) at 37 °C in 5% CO₂ environment.

For bioluminescence assays, 293T cells were seeded at a density of $2 \times 10^4$ cells per well in white opaque 96-well plates. At 24 h after plating, cells were treated with 10 μl of formulations containing 200 ng of FLuc mRNA (Vazyme) and incubated for another 24 h at 37 °C. Cells were lysed and assayed for luciferase activity with One-Lum Firefly Luciferase Reporter Gene Assay Kit (Beyotime) according to manufacturer's instructions.

For fluorescence imaging and flow cytometric assays, 293T ($2 \times 10^4$ cells per well) and THP-1 cells ($4 \times 10^4$ cells per well) were seeded in transparent 96-well plates for overnight culture. Cells were treated with ecoLNPs containing various amounts of eGFP mRNA (APExBio) or 200 ng of mCherry mRNA (Vazyme). At the designated time points, the fluorescence signals from cells were detected by fluorescence inverted microscopes (Nikon ECLIPSE Ts2R or Zeiss Axio Vert A1) and/or flow cytometer (Agilent NovoCyte 2000R). For three-dimensional cell culture, 293T cells ($1 \times 10^3$ cells per well) were seeded in a 96-well round-bottom plate coated with 3D cell culture coating solution (Beyotime) and treated with 50 ng of ecoLNPs-encapsulated eGFP mRNA, mCherry mRNA, or their combinations (25 ng plus 25 ng). In the case of flow cytometry, 10,000 events were analyzed by the NovoExpress software (Agilent).

For in situ β-Gal staining and colorimetric β-gal assays, 293T cells were seeded at a density of $2 \times 10^4$ cells per well in transparent 96-well plates. At 24 h after plating, cells were treated with ecoLNPs containing 200 ng of β-gal mRNA (TriLink). Approximately 24 h post-treatment, cells were imaged by a fluorescence inverted microscope (Nikon, ECLIPSE Ts2R) or β-gal activity was determined by a colorimetric assay kit (Beyotime)[48].

## siRNA and plasmid DNA delivery in vitro

For siRNA delivery, 293T-Luc2-tdT cells were seeded at a density of $2 \times 10^4$ cells per well in white opaque 96-well plates. At 24 h after plating,

ecoLNPs encapsulating siRNA (GAAGUGCUCGUCCUCGUCCdTdT, GenScript) targeting the luciferase reporter were prepared according to the above mRNA formulation protocol except that the weight ratio of CF3-2N6-UC18 to siRNA was 75:1 and added to each well for 48 h at 37 °C. Lipo2k (Thermo Fisher Scientific) was used as a positive transfection reagent according to manufacturer's instructions. Cells were lysed and assayed for luciferase activity with One-Lum Firefly Luciferase Reporter Gene Assay Kit (Beyotime) according to manufacturer's instructions. For plasmid DNA delivery, 293T ($2 \times 10^4$ cells per well) were seeded in transparent 96-well plates for overnight culture. Cells were treated with ecoLNPs containing 50 ng of pCMV-C-EGFP (Beyotime) for 24 h at 37 °C. The fluorescence signals from cells were detected by fluorescence inverted microscopes (Zeiss Axio Vert A1). EcoLNPs encapsulating pCMV-C-EGFP were prepared according to the above mRNA formulation protocol except that the weight ratio of CF3-2N6-UC18 to pCMV-C-EGFP was 8:1. Lipofectamine 3000 (Lipo3k, Thermo Fisher Scientific) was used as a positive transfection reagent according to manufacturer's instructions.

## Detection of serum or RNase A susceptibility of ecoLNPs-encapsulated mRNA

EcoLNPs containing 200 ng of mRNA were incubated with fetal bovine serum (FBS, Sigma) or RNase A (Sangon Biotech) at volume ratio of 9:1 for 6 h (final concentration, 10% and 50 μg ml⁻¹, respectively). Free mRNA incubated with the same concentration of FBS or 0.5 μg ml⁻¹ RNase A for 10 min served as a control. Samples were then mixed with loading buffer, loaded onto a 1% agarose gel for electrophoresis (120 V for 30 min), and imaged using a gel imaging system (Thermo Fisher Scientific, iBright FL1000).

## Hemolysis test

Hemolysis was carried out through incubating 10 μl of different concentrations of ecoLNPs with 90 μl of mouse red blood cells diluted in PBS adjusted to pH 7.4 or 5.4 (the final concentration of red blood cells was 2%, v/v) at 37 °C for 60 min. After centrifugation at $1000 \times g$ for 5 min, the amount of hemoglobin released into the supernatant was quantified by measuring the absorbance at 540 nm wavelength using a microplate reader (BioTek, Synergy LX). The absorbance values for the supernatant of cells treated with PBS and 1% Triton X-100 were used to define baseline and 100% hemolysis, respectively.

## Visualization of cellular uptake and endosomal escape

HeLa cells were seeded at a density of $1 \times 10^4$ cells per well in transparent 96-well plates. At 24 h after plating, cells were pre-incubated with PBS or various inhibitors including CPZ (Macklin), filipin (Meilunbio), EIPA (Aladdin), and Baf (Sangon Biotech) for 30 min at 37 °C (the final concentration was 10 μg ml⁻¹ for CPZ, 1 μg ml⁻¹ for filipin, 10 μg ml⁻¹ for EIPA, and 0.2 (or 1) μM for Baf, respectively)[49,50]. Cells were then treated with ecoLNPs containing 200 ng of Cy5-labeled mRNA (APExBIO) for 6 h at 37 °C or 4 °C, washed three times with PBS, and imaged using confocal fluorescence microscopy (Leica, TCS SP8). In a parallel experiment, cells subjected to the above procedures were resuspended in PBS for flow

cytometric analysis (Agilent, NovoCyte 2000R). To visualize endosomal escape of ecoLNPs, HeLa cells were first seeded in confocal dishes (bioshark) at a density of $1 \times 10^4$ cells per dish, incubated overnight at 37 °C, and treated with free Cy5 mRNA or ecoLNP-encapsulated Cy5 mRNA (final concentration of Cy5-labeled mRNA was 2 µg ml⁻¹) for 6 h at 37 °C. Cells were then stained with LysoTracker Green (Beyotime) for 45 min and Hoechst 33342 (Beyotime) for 10 min at 37 °C. Co-localization analysis was then acquired after washing by confocal fluorescence microscopy (Leica, TCS SP8)[51]. For calcein escape assay, cells were treated with ecoLNP-encapsulated Cy5-labeled mRNA (the final concentration of Cy5-labeled mRNA was 2 µg ml⁻¹) 3 h prior to addition of calcein (Sangon Biotech, the final concentration was 100 µg ml⁻¹). Cells were washed four times after incubation for another 3 h and imaged by confocal fluorescence microscopy (Leica, TCS SP8)[51]. LNPs prepared with PNT-2N6-UC18, a CF3-2N6-UC18 analog bearing a phenanthroline ring instead of a quinoline ring, were used as a control group throughout the endosomal escape assay.

### Titration of CF3-2N6-UC18

A titration of CF3-2N6-UC18 was carried out as described previously with slight modifications[33]. Briefly, 20 mg of CF3-2N6-UC18 dissolved in 1 ml of ethanol solution was mixed with 200 µl NaCl solution (1 mol l⁻¹), acidified with HCl (1 mol l⁻¹) to pH 1.5, adjusted to a final volume of 2 ml with deionized water, and titrated with NaOH (0.5 mol l⁻¹) at room temperature. The buffering capacity (BC) was calculated using the following equation: $BC = \Delta n \, V^{-1} \, \Delta pH^{-1}$, where $n$ and $V$ are the mole of base and the total volume, respectively. A titration of aqueous HCl was done using the same procedure.

### Measurement of endo-lysosomal pH

The endo-lysosomal pH was measured according to the previously reported protocol with slight modifications[34]. In brief, HeLa cells were first seeded in confocal dishes (bioshark) at a density of $1 \times 10^4$ cells per dish. After overnight incubation, the cells were incubated with 10 µl of PBS (or ecoLNPs containing 200 ng of FLuc mRNA), 250 µl of 10 kDa FITC-Dextran (2 mg ml⁻¹, Beyotime) dissolved in DPBS, and 250 µl of 10 kDa Alexa Fluor 647-Dextran (1 mg ml⁻¹, Thermo Fisher Scientific) dissolved in DPBS at 37 °C for 1 h. Following a washing step, cells were incubated for additional 2 h or 4 h and subjected to confocal fluorescence imaging (Leica, TCS SP8). In a parallel experiment, 10 kDa FITC-Dextran and 10 kDa Alexa Fluor 647-Dextran were added to the same final concentration to confocal dishes containing various buffers, with pH ranging from 4.5 to 7.5. The fluorescence intensity values of FITC-Dextran ($I_{FITC}$) and Alexa Fluor 647-Dextran ($I_{AF647}$) at different pH were measured at 488 nm and 647 nm using confocal fluorescence microscopy. A calibration curve was then generated using the ratio between $I_{FITC}$ and $I_{AF647}$ ($I_{FITC} \, I_{AF647}^{-1}$) and corresponding pH. The endo-lysosomal pH values of cells treated with PBS or ecoLNPs were calculated from the calibration curve.

### Molecular docking

The molecular docking of CF3-2N6-UC18 or PNT-2N6-UC18 on sapB was performed using AutoDock Vina (version 1.1.2)[52] according to the crystal structure of sapB in complex with chloroquine (PDB: 4V2O)[40]. Prior to docking, crystallographic waters were removed, polar hydrogen atoms were added, and the file was converted to pdbqt format using AutoDockTools (version 1.5.6). Meanwhile, lipids were energy minimized using ChemDraw. The grid box for CF3-2N6-UC18 and PNT-2N6-UC18 was set to include the active site pocket of sapB for chloroquine. The x, y, z dimensions of the grid box were settled as 22.5, 22.5, 22.5 Å, and the other parameters were as follows: x center, 7.086; y center, 83.922; z center, 92.552; spacing: 0.375 Å; and exhaustiveness: 10. The other parameters were kept at default values. Conformations with the lowest binding energy for each lipid were selected from nine conformations having top docking score for further analysis. Visualizations were generated using PyMOL (version 2.6.0a0).

### Fluorescence and bioluminescence imaging and safety evaluation in vivo

Animal care and experimental protocols were approved by the Institutional Animal Care and Use Committee of the First Affiliated Hospital of Southern University of Science and Technology in accordance with the guidelines for the care and use of laboratory animals. Mice were housed under standard conditions (12 h light and 12 h dark cycles, 20–22 °C, 40–60% humidity) with food and water provided ad libitum. C57BL/6 mice (female, 6–8 weeks old, Charles River) received a given single dose of ecoLNPs containing FLuc mRNA or Cy5-labeled mRNA through intravenous (IV), subcutaneous (SC), intramuscular (IM), subconjunctival (SCJ), or subretinal (SR) injection, or intravesical (IVS) instillation using a catheter inserted into the bladder. For SCJ and SR injections, as well as IVS instillation, mice were anesthetized during injections. In the case of IVS instillation, the urethral orifice was clamped with a clip for 1 h under anesthesia. At chosen measuring points, mice were anesthetized and placed in the imaging chamber for fluorescence or bioluminescence imaging using a small animal imaging system (PerkinElmer, Lumina LT). After whole-body imaging, mice were sacrificed, and the major tissues were excised for ex vivo imaging. For in vivo fluorescence imaging, the major tissues were also harvested for further frozen section analysis of the biodistribution of ecoLNPs. For in vivo bioluminescence imaging, luciferin potassium salt (Beyotime) was intraperitoneally injected into mice at a dose of 150 mg kg⁻¹ of body weight 10 min prior to imaging. For comparison, the same amount of FLuc mRNA was encapsulated into SM-102 LNPs (a delivery vehicle of FDA-approved COVID-19 mRNA vaccine, mRNA-1273) and intramuscularly injected into mice. The SM-102 formulation consisted of SM-102 (Macklin), DSPC (Avanti), Chol, and DMG-PEG2k (50:10:38.5:1.5, molar ratio) as well as the mRNA payloads (the weight ratio of SM-102 to mRNA was 10.75:1)[42,43]. For safety evaluation in vivo, the major tissues from mice were harvested at 4 h post-IV, IM, or SC injection of ecoLNPs at a mRNA dose of 0.5 mg kg⁻¹ for hematoxylin–eosin staining. Blood from mice receiving a single IV dose of ecoLNPs (equivalent to 0.5 mg kg⁻¹ of mRNA) was collected at a chosen measuring point for measurement of both liver and renal function as well as main serum biochemistry parameters.

### Cre mRNA- or CRISPR-Cas9 mRNA-mediated genome editing in vivo

For Cre mRNA-mediated genome editing in vivo, Ai9 tdTomato reporter mice (male, 6–9 weeks old, The Jackson Laboratory) were intramuscularly (hind legs) injected with ecoLNPs containing Cre mRNA (APExBio) at a mRNA dose of 0.5 mg kg⁻¹, respectively. Legs injected with PBS served as control groups. Mice were anesthetized 7 days post-injection for the detection of the tdTomato fluorescence signals using a small animal imaging system (PerkinElmer, Lumina LT). After whole-body imaging, the major tissues were excised for ex vivo imaging, frozen section analysis of the tdTomato fluorescence signals, and gel electrophoresis. For gel electrophoresis, DNA were extracted from tissues using FastPure Cell/Tissue DNA Isolation Mini Kit (Vazyme). After PCR amplification with specific primers (ATTTGA-TACCGCGGGCCCTAAG and GAACTCTTTGATGACCTCCTC)[53], samples were mixed with loading buffer, loaded onto a 1% agarose gel for electrophoresis (120 V for 30 min), and imaged using a gel imaging system (Thermo Fisher Scientific, iBright FL1000). In the case of CRISPR-Cas mRNA-mediated genome editing in vivo, Ai9 mice were intramuscularly injected with ecoLNPs containing 1 mg kg⁻¹ total RNA dose of Cas9 mRNA (APExBio) and gRNA (AAGTAAAACCTCTA-CAAATG, GenScript) targeting the loxP-flanked stop cassette locus at weight ratio of 3:1 on day 0 and day 7. On day 14, mice were anesthetized for the detection of tdTomato fluorescence signals at the whole-body level and euthanized for ex vivo imaging using a small animal imaging system (PerkinElmer, Lumina LT). Subsequently, genomic DNA was extracted from tissues using FastPure Cell/Tissue DNA

Isolation Mini Kit (Vazyme). Genomic segments spanning the sites of interest were amplified using a Phanta Max Master Mix (Vazyme) and specific primers (TCCTCCGGGCTGTAATTAGC and CCGAATTCGATC-TAGCTTGG). Following purification with a PCR Clean Up Kit (Beyo-time), PCR products were sequenced on a Sequel IIe system (PacBio).

## Statistical analysis

Data were presented as mean ± SEM. When relevant, the sample size (*n*) was provided in the corresponding figure legends. Statistical significance between two groups was analyzed using an unpaired, two-tailed Student's *t* test, and for more than two groups using one-way or two-way ANOVA with Tukey's or Sidak's multiple comparison test.

## Reporting summary

Further information on research design is available in the Nature Portfolio Reporting Summary linked to this article.

## Data availability

The raw sequencing data generated in this study have been deposited in the NCBI database under accession code BioProject ID PRJNA1240917. The crystal structure of chloroquine bound to sapB are available under PDB accession code 4V2O. Source data are available for Figs. 3a, 3d, 3e, 4c, 4d, 4h, 5a, 5d, 5f, 6b, 6c, and 6j and Supplementary Figs. 5a–c, 6a–c, 7a–e, 8b, 8d, 8g, 10a, 10d, 11b, 12a, 12c, 12d, 14b–d, and 15b in the associated source data file. Source data are provided with this paper.

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

## Acknowledgements

This work was supported by funding from the National Natural Science Foundation of China (82072053 to B.L.) and the Shenzhen Science and Technology Program (RCYX20200714114539061 to B.L.).

## Author contributions

B.L. conceived the study and supervised the project. Z.L. and J.W. performed the most of experiments and data analysis. N.W. assisted with lipid synthesis and animal experiments. Y.L., R.S., and M.Z. assisted with animal experiments. B.L. and Z.L. wrote, finalized, and corrected the manuscript. All authors provided critical feedback on the research, analysis and manuscript.

## Competing interests

B.L., Z.L., J.W., and N.W. are listed as inventors on a patent application describing the use of chloroquine-like lipid nanoparticles as a platform to deliver mRNA and other payloads (PCT/CN2025/078930). The other authors declare no competing interests.
