## [Transparent Peer Review file · Nature Communications]

Structure-guided design of endosomolytic chloroquine-like lipid nanoparticles for mRNA delivery and genome editing

Corresponding Author: Professor Bin Li

Version 0:

Reviewer comments:

Reviewer #1

(Remarks to the Author)

This manuscript, entitled "Structure-guided design of endosomolytic chloroquine-like lipid nanoparticles for mRNA delivery," presents a comprehensive approach for designing lipid nanoparticles (LNPs) optimized for mRNA delivery, leveraging structural insights from chloroquine-like compounds.

This study presents a novel lipid nanoparticle (LNP) platform, termed ecoLNPs, designed to enhance mRNA delivery through improved endosomal escape, addressing a key barrier in RNA therapeutics. Inspired by the endosomolytic properties of chloroquine, the authors developed a library of chloroquine-like lipids, optimized their formulations, and demonstrated superior mRNA delivery across various cell types compared to commercial transfection agents. Mechanistically, ecoLNPs leverage the proton sponge effect and possibly interact with saposin B to facilitate endosomal disruption, enabling efficient cytoplasmic release of mRNA. Using animal models *in vivo*, ecoLNPs exhibited effective mRNA and CRISPR-Cas9 delivery, with promising gene editing results and minimal toxicity, underscoring their potential for therapeutic applications in gene editing and mRNA-based treatments. The findings open new possibilities and contribute an innovative strategy for overcoming endosomal escape, offering significant implications for mRNA and genome editing technologies. The rational design of ecoLNPs tailored for pH-responsive escape is a valuable contribution to the development of more efficient delivery vehicles.

After careful consideration, we are unable to support the research work presented in this manuscript to warrant its recommendation for publication in the Nature Communications. The authors need to address a few key and major questions and comments would further enhance the manuscript's impact, and we kindly request the authors to do so prior to acceptance.

1. The abstract is generally clear, but it could benefit from more quantifiable details regarding the improvements ecoLNPs provide over standard formulations (e.g., specific fold increases in delivery efficiency or endosomal escape rates).
2. The manuscript details a methodical approach to the synthesis, optimization, and testing of lipid formulations, which is commendable. However, the dense presentation in some sections (Particularly the Figure 1 overview section and the optimization of formulations) may benefit from a more concise summary or flow chart depicting the optimization pipeline.
3. Reviewing their hydrophobic tails within their developed ionizable lipids shows good structural activity relationship studies. Numerous studies have shown that cationic and ionizable cationic lipids are broadly applied as auxiliary agents, but their use is associated with adverse effects. If these excipients are rapidly degraded to endogenously occurring metabolites such as amino acids and fatty acids, their toxic potential can be minimized.
 - <https://doi.org/10.1002/sml.202206968>
 - <https://pubs.acs.org/doi/10.1021/jacs.3c09143>
 - <https://doi.org/10.1038/s41467-021-27493-0>

For this impactful work, we would like to see the authors address the possible inflammatory responses and biodegradability of their best lead candidate using hydrolysis assays, enzyme-mediated degradation etc.

4. Although CF3-2N6-UC18 ecoLNPs maintain stability for at least a week in refrigerated conditions, ensuring stability

across varied physiological conditions remains a challenge. Degradation in vivo or under different storage conditions could affect therapeutic efficacy. We hope to see other routine conditions as well.

5. Mechanistically, the manuscript infers the proton sponge effect mainly through indirect evidence, such as the structural design of the ecoLNPs to include chloroquine-like protonatable groups and increased hemolytic activity under acidic endosomal pH. These properties suggest that ecoLNPs can absorb protons in acidic environments, leading to osmotic swelling and potential endosomal rupture.

Unlike more direct studies that measure endosomal pH changes or ion accumulation (such as via fluorescence-based pH sensors or ion-sensitive dyes), this manuscript does not provide direct measurements of pH or osmotic pressure within endosomes following ecoLNP uptake. In many existing studies, the proton sponge effect is confirmed by monitoring pH changes or by using specific inhibitors that block proton pumps to observe the effect on endosomal escape efficiency.

• <https://pubs.acs.org/doi/abs/10.1021/acsnano.7b07583>

• <https://link.springer.com/article/10.1007/s11095-022-03206-0>

• [https://www.cell.com/molecular-therapy-family/molecular-therapy/fulltext/S1525-0016\(16\)30570-6](https://www.cell.com/molecular-therapy-family/molecular-therapy/fulltext/S1525-0016(16)30570-6)

6. Ensure consistent terminology throughout the manuscript, particularly with terms like "ecoLNPs" and "endosomolytic nanoparticles." Variations in nomenclature could confuse readers at times. See conclusion section.

Consistent Terminology: Maintain consistent naming conventions for lipids (e.g., CF3-2N6-UC18) across figures, tables, and the main text to avoid confusion.

7. Discussing possible limitations of their developed ionizable lipids is crucial for a high-impact audience. Address potential scalability issues and any observed limitations in targeting specificity or immune response. Future directions, such as exploring other chloroquine-like compounds or fine-tuning particle size for specific applications, would add depth to the paper and demonstrate foresight.

8. The SI document includes a table of contents but could benefit from more descriptive headings for each section (e.g., "Synthesis of Chloroquine-like Lipids" instead of "Methods"). Detailed subheadings in the contents would also make navigation easier.

9. Similarly, while the synthetic methods are comprehensive, adding more intermediate steps or reaction conditions (e.g., temperature, duration for each specific reaction in multi-step synthesis) would aid reproducibility. Some figure legends (e.g., Figure S8 on surface charge and stability) are minimalistic. More comprehensive explanations for each panel within figures would make the document more user-friendly.

10. This manuscript includes a Spearman's correlation matrix for delivery efficiency and particle characteristics (i.e., Figure S7). Expanding on this with an interpretation of correlations, such as whether certain characteristics predict delivery efficiency, could add value and relevance to the supporting data.

11. For data analysis and statistical reporting on the in vitro and in vivo experiments (e.g., Figure S8), ensure consistent use of statistical annotations (e.g., p-values) and define these in figure legends. It would be helpful to specify the exact tests applied (e.g., two-tailed t-tests or ANOVA) and mention sample sizes (n values) for each condition.

Reviewer #2

(Remarks to the Author)

Z. Liu and coworkers developed a modular designed lipid for efficient deliver and escape of mRNA cargo. They screened the chemical structure for toxicity and functionality and settled on the CF3-2N6-UC18 structure and the best performer. This compound was tested in cell lines in 2D and 3D culture systems and evaluated for escape from endosomes into cytoplasm based mostly on confocal imaging. The animal studies were logically conceived and executed. What is really noteworthy here is that the authors developed a new delivery lipid-based compound based on a known chemical that facilitates endosomal escape. Endosomal escape is the big hurdle for getting cargo into the cytoplasm of cells. There are just a few minor clarifications that need to be added, but overall, this paper is quite good and relevant for vaccine delivery systems. Lines 121-123: Please indicate the cell lines that for expression.

Fig. 3e: Please explain why the optimal UC18 obtained the best performance, whereas, C12 and C17 were not determined? The text says that these substitutions had a reversed effect, so what does this mean?

Fig. 4b: What dose was used for transfection?

Fig. 4c: How do you know that a small RNase A protein did not degrade any of the mRNA within the encapsulated LNP? Were they lysed before loading in the agarose gel? Some arrows pointed to the parts of the gel where the RNA is located would be helpful. I am assuming the smudge near the bottom of the first lane is RNA.

Extended Fig. 5a: Images are too dark to assess. Can this data be quantified and graphed? Or is this Fig. 5a? If so, then modify the text to reflect this.

Fig. 5a: Bafilomycin A1 is attributed to blocking entry of the ecoLNP into cells. Would an alternative interpretation be that Baf

does not block internalization, but blocks efficient endosomal escape due to the lack of endosomal acidification?
Fig. 5c-f: The text of the manuscript needs improved wording to reflect this and to mention how many cell images were taken as 1-2 cells is not enough to base any conclusions.
Supplement Table 5: The AST levels were double for the LNP than PBS ctrl, but the SEM was also very high. Please double check those numbers and also determine if they are significantly different from the control.
All images. Please include sample sizes in all figure legends where relevant. This was lacking throughout the paper.
The mouse experiments were performed quite well showing localization, biodistribution, and efficacy of the mRNA cargo.
Methods: Detailed enough for duplication of experiments.
References are sufficient.

Version 1:

Reviewer comments:

Reviewer #1

(Remarks to the Author)

the authors have addressed my concerns and I now recommend publication

Reviewer #2

(Remarks to the Author)

The authors addressed all my comments and concerns and have made improvements with both the manuscript and accompanying figures. In my opinion, it is ready for publication.

We thank the reviewers for the constructive comments on our manuscript entitled "Structure-guided design of endosomolytic chloroquine-like lipid nanoparticles for mRNA delivery and genome editing" (manuscript ID: NCOMMS-24-83505-T). We have included a point-by-point response to address these comments. New revisions highlighted in red have strengthened the quality of our manuscript.

REVIEWER COMMENTS

Reviewer #1 (Remarks to the Author):

This manuscript, entitled "Structure-guided design of endosomolytic chloroquine-like lipid nanoparticles for mRNA delivery," presents a comprehensive approach for designing lipid nanoparticles (LNPs) optimized for mRNA delivery, leveraging structural insights from chloroquine-like compounds.

This study presents a novel lipid nanoparticle (LNP) platform, termed ecoLNPs, designed to enhance mRNA delivery through improved endosomal escape, addressing a key barrier in RNA therapeutics. Inspired by the endosomolytic properties of chloroquine, the authors developed a library of chloroquine-like lipids, optimized their formulations, and demonstrated superior mRNA delivery across various cell types compared to commercial transfection agents. Mechanistically, ecoLNPs leverage the proton sponge effect and possibly interact with saposin B to facilitate endosomal disruption, enabling efficient cytoplasmic release of mRNA. Using animal models *in vivo*, ecoLNPs exhibited effective mRNA and CRISPR-Cas9 delivery, with promising gene editing results and minimal toxicity, underscoring their potential for therapeutic applications in gene editing and mRNA-based treatments. The findings open new possibilities and contribute an innovative strategy for overcoming endosomal escape, offering significant implications for mRNA and genome editing technologies. The rational design of ecoLNPs tailored for pH-responsive escape is a valuable contribution to the development of more efficient delivery vehicles.

After careful consideration, we are unable to support the research work presented in this manuscript to warrant its recommendation for publication in the Nature Communications. The authors need to address a few key and major questions and comments would further

enhance the manuscript's impact, and we kindly request the authors to do so prior to acceptance.

We thank the reviewer for the positive feedback and comments. We have addressed each comment below.

1. The abstract is generally clear, but it could benefit from more quantifiable details regarding the improvements ecoLNPs provide over standard formulations (e.g., specific fold increases in delivery efficiency or endosomal escape rates).

We thank the reviewer for this suggestion to improve our abstract. A quantitative description on improvements in the delivery efficiency of ecoLNPs over standard controls has been included in the Abstract section (Page 2, line 18).

2. The manuscript details a methodical approach to the synthesis, optimization, and testing of lipid formulations, which is commendable. However, the dense presentation in some sections (Particularly the Figure 1 overview section and the optimization of formulations) may benefit from a more concise summary or flow chart depicting the optimization pipeline.

Figure 1 has been redrawn according to the reviewer's suggestion.

3. Reviewing their hydrophobic tails within their developed ionizable lipids shows good structural activity relationship studies. Numerous studies have shown that cationic and ionizable cationic lipids are broadly applied as auxiliary agents, but their use is associated with adverse effects. If these excipients are rapidly degraded to endogenously occurring metabolites such as amino acids and fatty acids, their toxic potential can be minimized.

- <https://doi.org/10.1002/smll.202206968>

- <https://pubs.acs.org/doi/10.1021/jacs.3c09143>

- <https://doi.org/10.1038/s41467-021-27493-0>

For this impactful work, we would like to see the authors address the possible inflammatory responses and biodegradability of their best lead candidate using hydrolysis assays, enzyme-mediated degradation etc.

We thank the reviewer for this comment and agree that it is critical to evaluate the possible adverse effects induced by the lead candidate. Aside from in vitro hemolysis testing (Fig. 4d), we also had performed in vivo safety evaluation of ecoLNPs in the original manuscript

including serum biochemical and haematological analyses on mouse serum (Supplementary Table 5) as well as examinations of pathological changes in major tissues following various administration routes (Supplementary Fig. 13).

In recent years, a variety of LNPs have been developed for mRNA delivery at an unprecedented speed. From a chemical structure perspective, the key cationic or ionizable lipids in these LNPs can be divided into biodegradable lipids and lipids without degradable bonds. While it is a common strategy to introduce degradable ester or disulfide bonds into lipids to increase their metabolic rate, numerous lipids without degradable bonds also hold great promise for multiple applications (Figure X) ¹⁻³. For instance, C12-200 LNPs have been widely used as a benchmark formulation for mRNA delivery;¹ C12-494 LNPs are capable of facilitating mRNA transport across the blood-brain barrier;² LNPs 55 enable targeted mRNA delivery to the placenta for treating pre-eclampsia.³ Given that CF3-2N6-UC18 does not contain biodegradable bonds, we think it is difficult to evaluate its metabolic rate using hydrolysis assays or enzyme-mediated degradation.

We are currently developing chloroquine-like lipid derivatives with various degradable bonds for future exploration, which is beyond the scope of this study. We have now included a brief discussion on designing degradable counterparts for future work (Page 26, line 468: As biodegradable lipids tend to degrade more rapidly than those without degradable bonds^{13, 46, 47}, future work will focus on designing biodegradable counterparts and comparing their efficacy and safety to lipids developed in this study).

Figure X. Chemical structures of lipids C12-200¹, 55², and C12-494³.

References:

1. Kauffman KJ, Dorkin JR, Yang JH, Heartlein MW, DeRosa F, Mir FF, Fenton OS, Anderson DG. Optimization of lipid nanoparticle formulations for mRNA delivery in vivo with fractional factorial and definitive screening designs. *Nano Lett.* **15**, 7300-7306 (2015).
2. Swingle KL, Hamilton AG, Safford HC, et al. Placenta-tropic VEGF mRNA lipid nanoparticles ameliorate murine pre-eclampsia. *Nature* **637**, 412-421 (2025).
3. Han EL, Padilla MS, Palanki R, et al. Predictive High-Throughput Platform for Dual Screening of mRNA Lipid Nanoparticle Blood-Brain Barrier Transfection and Crossing. *Nano Lett.* **24**, 1477-1486 (2024).
4. Although CF3-2N6-UC18 ecoLNPs maintain stability for at least a week in refrigerated conditions, ensuring stability across varied physiological conditions remains a challenge. Degradation in vivo or under different storage conditions could affect therapeutic efficacy. We hope to see other routine conditions as well.

To determine whether other routine conditions affect the stability of ecoLNPs, we stored them at room temperature (25 °C) and tested their changes in the particles size and delivery performance. New experiments indicated that a further increase in storage temperature posed a negative effect on formulations' delivery performance, albeit that no significant changes with time in the particle size occurred (Supplementary Figs. 8b and 8c). These new results included in the revised manuscript (Page 13, line 205) suggested that non-frozen cold storage was required to preserve formulations' stability and activity.

5. Mechanistically, the manuscript infers the proton sponge effect mainly through indirect evidence, such as the structural design of the ecoLNPs to include chloroquine-like protonatable groups and increased hemolytic activity under acidic endosomal pH. These properties suggest that ecoLNPs can absorb protons in acidic environments, leading to osmotic swelling and potential endosomal rupture.

Unlike more direct studies that measure endosomal pH changes or ion accumulation (such as via fluorescence-based pH sensors or ion-sensitive dyes), this manuscript does not provide direct measurements of pH or osmotic pressure within endosomes following ecoLNP uptake. In many existing studies, the proton sponge effect is confirmed by monitoring pH changes or by using specific inhibitors that block proton pumps to observe the effect on endosomal escape efficiency.

<https://pubs.acs.org/doi/abs/10.1021/acsnano.7b07583>

<https://link.springer.com/article/10.1007/s11095-022-03206-0>

[https://www.cell.com/molecular-therapy-family/molecular-therapy/fulltext/S1525-0016\(16\)30570-6](https://www.cell.com/molecular-therapy-family/molecular-therapy/fulltext/S1525-0016(16)30570-6)

In the original manuscript, we had utilized a specific V-type H⁺-ATPase inhibitor, Bafilomycin A1, to explore the effects of proton pumps on the cellular uptake of ecoLNPs. As this inhibitor dramatically blocks ecoLNPs entry into cells, we tried to confirm the proton sponge effect in the revised manuscript by measuring the buffering capacity of CF3-2N6-UC18 and monitoring endo-lysosomal pH changes following ecoLNP uptake using fluorescence-based pH sensors according to literatures provided by the reviewer. The high buffering capacity of CF3-2N6-UC18 at endo-lysosomal pH range calculated from its

titration curve (Supplementary Figs. 10a and 10b) indicated that it possesses the potential to serve as a proton sponge to absorb a substantial amount of protons. In addition, a significant increase in the endo-lysosomal pH following ecoLNP uptake was observed (Supplementary Figs. 10c and 10d), which was in line with the results from other reagents that function as proton sponges¹⁻³. These new data included in the revised manuscript (Page 13, line 205) provide direct evidence to support the proton sponge effect mediated by ecoLNPs.

References:

1. Sonawane, N.D., Szoka, F.C., Jr. & Verkman, A.S. Chloride accumulation and swelling in endosomes enhances DNA transfer by polyamine-DNA polyplexes. *J Biol. Chem.* **278**, 44826-44831 (2003).
2. Wilson, D.R. et al. A Triple-Fluorophore-Labeled Nucleic Acid pH Nanosensor to Investigate Non-viral Gene Delivery. *Mol. Ther.* **25**, 1697-1709 (2017).
3. Forrest, M.L. & Pack, D.W. On the kinetics of polyplex endocytic trafficking: implications for gene delivery vector design. *Mol. Ther.* **6**, 57-66 (2002).

6. Ensure consistent terminology throughout the manuscript, particularly with terms like "ecoLNPs" and "endosomolytic nanoparticles." Variations in nomenclature could confuse readers at times. See conclusion section.

Consistent Terminology: Maintain consistent naming conventions for lipids (e.g., CF3-2N6-UC18) across figures, tables, and the main text to avoid confusion.

We apologize for any confusion. The top-performing CF3-2N6-UC18 LNPs have been termed endosomolytic chloroquine-like optimized lipid nanoparticles (ecoLNPs) terminology throughout the revised manuscript including the main text, figures, tables, and supplementary data.

7. Discussing possible limitations of their developed ionizable lipids is crucial for a high-impact audience. Address potential scalability issues and any observed limitations in targeting specificity or immune response. Future directions, such as exploring other chloroquine-like compounds or fine-tuning particle size for specific applications, would add depth to the paper and demonstrate foresight.

We thank the reviewer for this suggestion and have now included a brief discussion on designing degradable counterparts for future work (Page 26, line 468: As biodegradable lipids tend to degrade more rapidly than those without degradable bonds^{13, 46, 47}, future work will focus on designing biodegradable counterparts and comparing their efficacy and safety to lipids developed in this study).

8. The SI document includes a table of contents but could benefit from more descriptive headings for each section (e.g., "Synthesis of Chloroquine-like Lipids" instead of "Methods"). Detailed subheadings in the contents would also make navigation easier.

As the revised SI document does not include a table of contents, more specific information can be found in the corresponding titles describing figures and tables.

9. Similarly, while the synthetic methods are comprehensive, adding more intermediate steps or reaction conditions (e.g., temperature, duration for each specific reaction in multi-step synthesis) would aid reproducibility. Some figure legends (e.g., Figure S8 on surface charge and stability) are minimalistic. More comprehensive explanations for each panel within figures would make the document more user-friendly.

1. In the original manuscript, we had provided a general but detailed procedure including temperature and duration for the synthesis of the lead lipid in the Method sections. In the revised manuscript, we have added the following descriptions: The other CIs were synthesized with their corresponding starting materials using the same procedure. For X-2Nz-Cn, X-3Nz-Cn, and X-SPN-Cn, the molar feed ratios of intermediate (X-yNz):alkyl bromide (Cn):K₂CO₃:KI were 1:2.4:2.5:0.5, 1:3.6:3.7:0.5, and 1:4.8:4.9:0.5, respectively (Page 28, line 530).

2. More detailed explanations for some panels have been included in the revised manuscript.

10. This manuscript includes a Spearman's correlation matrix for delivery efficiency and particle characteristics (i.e., Figure S7). Expanding on this with an interpretation of correlations, such as whether certain characteristics predict delivery efficiency, could add value and relevance to the supporting data.

An interpretation of Spearman's correlation matrix (Page 9, line 169: This result suggested

that the physicochemical parameters examined in this study were not ideal indices to predict formulations' mRNA delivery efficiency) has been added in the revised manuscript.

11. For data analysis and statistical reporting on the in vitro and in vivo experiments (e.g., Figure S8), ensure consistent use of statistical annotations (e.g., p-values) and define these in figure legends. It would be helpful to specify the exact tests applied (e.g., two-tailed t-tests or ANOVA) and mention sample sizes (n values) for each condition.

Statistical information has been included in the figure legends and the Methods (Statistical Analysis) section.

Reviewer #2 (Remarks to the Author):

Z. Liu and coworkers developed a modular designed lipid for efficient deliver and escape of mRNA cargo. They screened the chemical structure for toxicity and functionality and settled on the CF3-2N6-UC18 structure and the best performer. This compound was tested in cell lines in 2D and 3D culture systems and evaluated for escape from endosomes into cytoplasm based mostly on confocal imaging. The animal studies were logically conceived and executed. What is really noteworthy here is that the authors developed a new delivery lipid-based compound based on a known chemical that facilitates endosomal escape. Endosomal escape is the big hurdle for getting cargo into the cytoplasm of cells. There are just a few minor clarifications that need to be added, but overall, this paper is quite good and relevant for vaccine delivery systems.

We thank the reviewer for the positive feedback and comments. We have addressed each comment below.

Lines 121-123: Please indicate the cell lines that for expression.

The cell line for FLuc expression has been provided in the revised manuscript (Page 8, line 124).

Fig. 3e: Please explain why the optimal UC18 obtained the best performance, whereas, C12 and C17 were not determined? The text says that these substitutions had a reversed effect, so what does this mean?

Since the majority of X-3N6-Cn with tails of C12 and C17 showed decreased mRNA

delivery efficiency compared with their corresponding UC18 tails, we did not synthesize X-2N4-Cn and X-2N6-Cn with tails of C12 and C17.

Regarding the sentence “these substitutions had a reversed effect”, substitutions herein indicate the substituent groups in the quinoline ring (scaffold) rather than the C12, C17, and UC18 tails. To avoid confusion, we have specified the substitution position in the revised manuscript (Page 9, line 158).

Fig. 4b: What dose was used for transfection?

The dose used for transfection has been provided in the figure legend of Fig. 4b (Page 15, line 214).

Fig. 4c: How do you know that a small RNase A protein did not degrade any of the mRNA within the encapsulated LNP? Were they lysed before loading in the agarose gel? Some arrows pointed to the parts of the gel where the RNA is located would be helpful. I am assuming the smudge near the bottom of the first lane is RNA.

1. In comparison to ecoLNPs-encapsulated mRNA without RNase A treatment (lane 4), incubation of ecoLNPs-encapsulated mRNA with RNase A for 10 min (lane 7) and 6 h (lane 8) did not lead to obvious changes in the intensity of bands. We thus concluded that ecoLNPs can protect mRNA from RNase A-mediated degradation.
2. Samples were not lysed before loading in the agarose gel.
3. The dark band near the bottom (lane 1) indicates free mRNA.
4. The arrows pointed to the free mRNA and ecoLNPs-encapsulated mRNA have been provided in Fig. 4c.

Extended Fig. 5a: Images are too dark to assess. Can this data be quantified and graphed? Or is this Fig. 5a? If so, then modify the text to reflect this.

We apologize for any confusion. We used both confocal microscopy (qualitative analysis, Supplementary Fig. 9a) and flow cytometry (quantitative analysis, Supplementary Fig. 9b) to investigate the internalization of ecoLNP. In addition, we also used the latter to quantify the effects of specific inhibitors of endocytic pathways on cellular entry of ecoLNPs. (Fig. 5a). To avoid confusion, we have updated this part in the revised manuscript (Page 17, line 276: We subsequently investigated their internalization process using both confocal

microscopy and flow cytometry and found that ecoLNPs were internalized by cells within hours (Supplementary Figs. 9a and 9b)).

Fig. 5a: Bafilomycin A1 is attributed to blocking entry of the ecoLNP into cells. Would an alternative interpretation be that Baf does not block internalization, but blocks efficient endosomal escape due to the lack of endosomal acidification?

We thank the reviewer for pointing out this interesting question. If Bafilomycin A1 only blocks efficient endosomal escape instead of internalization, it will cause substantial Cy5 signals (red) entrapped inside the endosomal which should still be detected by both confocal microscopy (Supplementary Fig. 9a) and flow cytometry (Fig. 5a). However, only weak red fluorescence could be detected in ecoLNPs-treated cells pretreated with either high (1 μ M) or low (0.2 μ M) dose of Bafilomycin A1 compared with the cells without any pretreatment (Supplementary Fig. 9a and Fig. 5a). Thus, it is reasonable to infer that Bafilomycin A1 blocks ecoLNPs entry into cells.

We do not rule out the possibility that Bafilomycin A1 simultaneously affects cellular internalization and endosomal escape of ecoLNPs. If so, it will also provide further proof for the endosomolytic action of ecoLNPs. We therefore have updated this section to include discussion on the possible effect of Bafilomycin A1 on ecoLNPs-mediated endosomal escape in the revised manuscript (Page 19, line 315: The result was in agreement with the previous observation³¹ and could plausibly be attributed to the blockade of CF3-2N6-UC18 ecoLNPs transport from endosomes to lysosomes or escape from endosomes and subsequent disturbance in endocytosis and intracellular trafficking; Page 20, line 321: As CF3-2N6-UC18 serves as a weak Brønsted base capable of accepting a proton from acidic conditions and its internalization as well as endosomal escape can be blocked by the proton pump inhibitor Baf, it is reasonable to infer that the endosomolytic action of ecoLNPs is relevant to the proton sponge effect, a hypothesis to explain endosomal escape of vehicles with the large buffering capacities via buffering endo-lysosomes, increasing the influx of protons, and causing the rupture of endo-lysosomal membranes).

Fig. 5c-f: The text of the manuscript needs improved wording to reflect this and to mention how many cell images were taken as 1-2 cells is not enough to base any conclusions.

Actually, over a dozen of images were taken for each sample and the most representative image was presented. We have added such information in the figure legend of Figs. 5c and 5e (Page 19, line 294: Over a dozen of images were taken for each sample and the most representative image was presented).

Supplement Table 5: The AST levels were double for the LNP than PBS ctrl, but the SEM was also very high. Please double check those numbers and also determine if they are significantly different from the control.

We have rechecked the AST levels carefully. No statistically significant differences were observed for the AST level and all other parameters tested. We have included statistical analysis in the revised Supplementary Table 5.

All images. Please include sample sizes in all figure legends where relevant. This was lacking throughout the paper.

Statistical information including sample sizes has been included throughout the revised manuscript.

The mouse experiments were performed quite well showing localization, biodistribution, and efficacy of the mRNA cargo. Methods: Detailed enough for duplication of experiments. References are sufficient.

We thank the reviewer for the positive feedback and comments.